# Archive of bacterial community in anhydrite crystals from a deep-sea basin provides evidence of past oil-spilling in a benthic environment in the Red Sea

Yong Wang[1,2], Tie Gang Li[3,4], Meng Ying Wang[1], Qi Liang Lai[5], Jiang Tao Li[6], Zhao Ming Gao[1], Zong Ze Shao[5], Pei-Yuan Qian[2,*]

[1]Institute of Deep-Sea Science and Engineering, Chinese Academy of Sciences, San Ya, China

[2] Division of Life Science, Hong Kong University of Science and Technology, Clear Water Bay, Hong Kong, China

[3] Key Laboratory of Marine Sedimentology and Environmental Geology, First Institute of Oceanography, State of Oceanic Administration (SOA), Qingdao, China

[4]Laboratory for Marine Geology, Qingdao National Laboratory for Marine Science and Technology, Qingdao, China

[5] Key Laboratory of Marine Biogenetic Resources, The Third Institute of Oceanography, SOA, Xiamen, China

[6] State Key Laboratory of Marine Geology, Tongji University, Shanghai, China

**Corresponding author:** boqianpy@ust.hk, Tel: 852-2358-7331

**Keywords:** *Alcanivorax*; metagenome; anhydrite; Atlantis II brine pool; hydrothermal sediment

**Running title:** Archive of microbial inhabitants in anhydrites

**Abstract**
In deep-sea sediment, the microbes present in anhydrite crystals are potential markers of
the past environment. In the Atlantis II Deep, anhydrite veins were produced by mild
mixture of calcium-rich hydrothermal solutions and sulfate in the bottom water, which
had probably preserved microbial inhabitants in the past seafloor of the Red Sea. In this
study, this hypothesis was tested by analyzing the metagenome of an anhydrite crystal
sample from the Atlantis II Deep. The estimated age of the anhydrite layer was between
750-770 years, which might span the event of hydrothermal eruption into the benthic
floor. The 16S/18S rRNA genes in the metagenome were assigned to Bacteria, Archaea,
Fungi and even invertebrate species. The dominant species in the crystals was an
oil-degrading *Alcanivorax borkumensis* bacterium, which was not detected in the adjacent
sediment layer. Fluorescence microscopy using 16S rRNA and marker gene probes
revealed intact cells of the *Alcanivorax* bacterium in the crystals. A draft genome of *A.*
*borkumensis* was binned from the metagenome. It contained all functional genes for
alkane utilization and the reduction of nitrogen oxides. Moreover, the metagenomes of
the anhydrites and control sediment contained aromatic degradation pathways, which
were mostly derived from *Ochrobactrum* sp. Altogether, these results indicate an oxic,
oil-spilling benthic environment in the Atlantis II basin of the Red Sea in approximately
the 14th century. The original microbial inhabitants probably underwent a dramatic
selection process via drastic environmental changes following the formation of an
overlying anoxic brine pool in the basin due to hydrothermal activities.



# 1. Introduction

Deep-sea sediment is among the least explored biospheres on Earth. Indigenous microbes differ vastly in community composition and metabolic spectra at different depths and sites (Orcutt et al., 2011;Teske and Sorensen, 2007). The distribution of microbes in subsuperficial sediments is determined by the porosity, nutrient availability and geochemical conditions of the sediment (Parkes et al., 2000;Webster et al., 2006). In return, genomic features and the community composition of the indigenous microbial inhabitants may reflect the *in situ* conditions and serve as biomarkers containing the geochemical indicators. However, most of the biomarkers cannot be well preserved and will be degraded by biological and abiological activities. Although lipids and other organic carbons present in some minerals allow the interpretation of microbial activities to some extent (Brocks et al., 2005), the original metabolic activities are difficult to retrieve in a comprehensive and precise manner.

Most of the dead microbes are damaged during the sedimentation process, but some can be maintained in almost their original shape (Taher, 2014;Benison et al., 2008). Evaporites, which mostly consist of halite and anhydrite ($CaSO_4$) or gypsum ($CaSO_4 \cdot 2H_2O$, temperature <38℃ (Hill, 1937)), are common microbialites with accretionary organosedimentary structures (Dupraz et al., 2011). Numerous dead bacteria, algae and metazoans have been detected in gypsum granules (Petrash et al., 2012;Trichet et al., 2001); bacterial mats growing on evaporites may become trapped and constitute much larger microbialites (Babel, 2004). Consequently, microbial inhabitants on the benthic surface may get trapped in the evaporites (Benison et al., 2008). Anhydrite facies are not found throughout deep-sea sediments. They usually form around hydrothermal vents in deep-sea environments (Jannasch and Mottl, 1985). A strong deep-sea volcanic eruption may break the crustal basalts, resulting in a drastic emission of hydrothermal gases followed by the crystallization of anhydrites and the deposition of metal sulfides (Kristall et al., 2006). An alternative model is that mild hydrothermal activities lead to a slow influx of solutions into the overlying sediment at temperatures in the sub-seafloor ranging from 20-100℃. This process also results in the formation of crystalline anhydrites in veins and around warm vents (Jannasch and Mottl, 1985). The latter process

may trap microbial inhabitants on the seafloor and within surface layers in anhydrites.
Due to the mild temperature, the trapped bodies are better preserved as excellent
biological evidence for past geochemical conditions.

A similar mild hydrothermal field is present in the Red Sea. Initially found in a deep-sea
rift in the 1960s (Swallow and Crease, 1965;Girdler, 1970), the temperature of the
Atlantis II brine pool has recently increased to 68°C (Anschutz and Blanc, 1996). In 1972,
several sediment cores were obtained from the southwest region of the pool (DSDP
Site226), and metal sulfides and evaporites were recognized as major mineral facies in
this brine-filled basin (Shipboard Scientific Party, 1974). In particular, thick and
well-crystalized anhydrite layers were found within the hematite and at the bottom of the
cores. Two major anhydrite units were later defined by analysis of the adjacent core
samples. The lower unit comprised anhydrite ranging from 12 to 70 wt% (Anschutz et al.,
2000). The anhydrite in the sediments likely resulted from a geyser-type eruption of
hydrothermal solutions into the Atlantis II brine pool followed by the mixing of
calcium-rich solutions with dissolved sulfate-bearing brine and the precipitation of
anhydrites during the cooling process (Ramboz et al., 1988). The discovery of veins
containing sulfides and anhydrite in the sediment suggests that a mild hydrothermal
eruption created the anhydrite facies in the Atlantis II sediment (Zierenberg and Shanks,
1983;Oudin et al., 1984;Missack et al., 1989). The formation of anhydrite facies in this
manner would trap microbial cells and organic debris in the bottom water and surface
sediment. These anhydrite layers probably contained important indigenous microbial
inhabitants during the occurrence of the hydrothermal events at the deep-sea benthic floor
of the Red Sea. Coupled with $^{14}$C markers to estimate age, the anhydrite facies contained
a large quantity of information regarding the past geochemical changes. The formation
process of the Atlantis II brine pool is still controversial, largely because the source of the
brine is uncertain (Schardt, 2016). The brine water had converted the bottom of the deep
into anoxic, hypersaline and hot environment. The microbes in the anhydrite facies may
provide hints for the original benthic conditions and age of the pool. It is also an
interesting question whether oil was generated in the sediments under the mild
hydrothermal activities in the past deep. If yes, seeping hydrothermal solutions may bring
oil into the seafloor of the deep, which might be documented by the microbes in the
anhydrites.
In the present study, we sampled a sediment core near Site226 and detected an anhydrite
layer. The dominant species were alkane- and oil-degrading bacteria indicating an oxic,
oil-spilling benthic condition when the layer was formed. The present study sheds light
on the importance of anhydrites in deep-sea sediment as an archive of microbial
inhabitants that can serve as biomarkers of past geochemical events.

**2     Materials and methods**
**2.1 Physicochemical measurements of sediment layers**
In 2008, a 2.25-meter gravity sediment core was obtained from the southwest basin
(approximately 2180 meters below sea level) of the Atlantis II Deep (21°20.76' N,
38°04.68' E) in the Red Sea (Fig. S1) (Bower, 2009). The core was frozen at -80°C and
then sliced aseptically into seventy-five 3-cm sections. Microbes from sediment slices of
12-15 cm, 63-66 cm, 105-108 cm, 183-186 cm, and 222-225 cm were first suspended in
phosphate-buffered saline and shaken on a vortexer for 30 s. After 30 minutes, the
supernatant was filtered through a 0.22-μm black polycarbonate filter. After
6-diamidino-2-phenyloindole (DAPI) staining, the microbes from each layer were
counted under an epifluorescence microscope (n = 3) (Gough and Stahl, 2003). The pore
water from the above 5 layers was collected by centrifugation. The concentration of
dissolved organic carbon (DOC) in the pore water was determined using the combustion
method (Trichet et al., 2001). The concentrations of ammonium, nitrite and nitrate were
measured using a TNM-I analyzer (Simadzu, Kyoto, Japan). To separate large particles
(>63 μm) from small particles (<63 μm), the sediment samples were passed through a
63-μm stainless steel sieve. The percentage of small particles (dry weight) was calculated
for all slices.

The age of the sections was estimated with a radiometric dating method that utilizes the
naturally occurring radioisotope $^{14}$C. The monospecific *Globigerinoides sacculifer*
specimens ranging in size from 250 to 350 μm were manually selected with caution and
then subjected to $^{14}$C measurement in the National Ocean Sciences Accelerator Mass
Spectrometry (AMS) Facility at the Woods Hole Oceanographic Institute, USA. The raw
AMS $^{14}$C ages were converted to calendar ages using the CALIB 6.0 program
(http://calib.qub.ac.uk/calib/) with the dataset Marine 09 (Reimer et al., 2009). A
reservoir correction has been considered for the $^{14}$C difference between atmospheric and
surface waters (Bard, 1988).
**2.2 DNA extraction and amplification**
The boundary of the anhydrite layer was determined by naked eye observation and
particle size measurement. Crystals were manually collected from the layers, followed by
ultrasonic cleaning. The homogenized crystals were then analyzed by X-ray
diffraction (XRD) (Rigaku, Tokyo, Japan) using Cu K-alpha radiation of 40 kV and 30
mA. The following procedure was conducted for DNA extraction from the crystals with
caution to avoid contamination. Surface contamination was removed by rinsing with 70%
alcohol in autoclaved distilled deionized water, followed by pulsed ultrasonic cleaning
for 2 hours. Anhydrite crystals (20 g) (Fig. 1A) of different sizes were treated with 1 μL
(2U) Turbo DNase I (Ambion, Austin, Texas, US) for 30 m in a 37°C incubation before
being ground for DNA extraction in a sterile hood. The anhydrite powder was used for
DNA extraction with the PowerSoil DNA Isolation kit (MO-BIO, Carlsbad, USA),
followed by a purification step according to the manufacturer's instructions. The DNA
concentration was quantified with a Quant-iT PicoGreen kit (Invitrogen, USA). Twenty
picograms of the raw DNA extract was used for DNA amplification using a MALBAC
kit (Yikang, Jiangsu, China) according to the manufacturer's manual (Zong et al., 2012).
The MALBAC amplification method has been evaluated recently in metagenomic studies
(Wang et al., 2016). Two MALBAC amplification assays were conducted using
twenty-one PCR cycles to acquire a sufficient amount of DNA for subsequent sequencing.
A negative control was also incorporated in the assay. The DNA concentration of the
MALBAC-amplified sample and the negative control was measured with a Bioanalyzer
(Agilent, CA, US). The products of the MALBAC amplification and negative control
were examined by gel electrophoresis to confirm the size ranges of the amplicons. Three
replicates of MALBAC amplifications for each sample were mixed and used for Illumina
sequencing on a Hiseq2000 platform (Illumina, San Diego, US). As a control, 10 g of
sediment from a position at 168 cm from the top of the core was used for DNA extraction.
There were no recognizable anhydrite crystals in this layer. DNA sequencing was
conducted as described above.

**2.2 Binning of metagenomes**
The initial Illumina 2×110-bp paired-end reads were subjected to quality assessments
using the NGS QC Toolkit with default parameters (Patel and Jain, 2012). The Illumina
sequencing data were deposited in the NCBI SRA database (accession number
SRA356974). The 35-bp MALBAC adapters at the two ends of the sequencing reads
were removed. Assembly of the trimmed Illumina 2×75-bp paired-end reads was
performed using SPAdes 3.5 (Nurk et al., 2013). The read coverage for the assembled
contigs was calculated using SAMtools (Li et al., 2009). The 16S/18S rRNA genes in the
contigs were identified using rRNA_HMM (Huang et al., 2009). Using classify.seqs
command in mothur package (Schloss et al., 2009), taxonomic sorting of the 16S rRNA
genes was conducted against the SILVA database with a confidence threshold of 80%.
The relative abundance of the species in the metagenomes was roughly estimated based
on the coverage of the 16S/18S rRNA genes. Binning of the draft genomes was
performed based on the read coverage and G+C content of the contigs (Fig. 1B), followed
by principal component analysis (PCA) of the tetranucleotide frequencies (TNF) of their
respective contigs using a previously described pipeline (Fig. 1C) (Albertsen et al., 2013).
The R scripts (R Core Team, 2013) for the binning process were obtained from
https://github.com/MadsAlbertsen/multi-metagenome. To evaluate the completeness of
the draft genome, conserved single-copy genes (CSCGs) were counted in the genome.
The CSCGs were identified by searching the CDSs against a database of essential
bacterial genes (107 essential genes) (Albertsen et al., 2013) using hmmsearch (3.0) with
default cutoffs for each protein family (Krogh et al., 1994).

**2.3 Genomic analyses**
The coding DNA sequences (CDSs) of the draft genome were predicted using Prodigal
(version 2.60)(Hyatt et al., 2010). KEGG annotation of the CDSs was performed using
BLASTp against the KEGG database (Kanehisa et al., 2012) with a maximum e-value
cutoff of 1e-05. The KEGG pathways were reconstructed using the KEGG website
(http://www.kegg.jp). CDSs were also annotated against the NCBI NR database, and
MEGAN was used for taxonomic affiliation and SEED/subsystem annotation of the
CDSs (Overbeek et al., 2005). The draft genome was submitted to NCBI (accession
number LKAP00000000). The average nucleotide identity (ANI) was calculated using
the algorithm integrated in the web service of EZGenome (Goris et al., 2007). The
DNA-DNA hybridization (DDH) estimate value was calculated using the
genome-to-genome distance calculator (GGDC2.0) (Meier-Kolthoff et al., 2013;Auch et
al., 2010a;Auch et al., 2010b).

**2.4 Detection and phylogeny of 16S ribosomal RNA (rRNA) genes**
The 16S rRNA gene sequence was identified from the draft genome sequence. The
closest relatives based on 16S sequence similarity were determined using the web service
of EzTaxon (Kim et al., 2012). The neighbour-joining phylogenetic tree was constructed
using MEGA version 5.0 (Tamura et al., 2011) with the Kimura 2-parameter model. The
phylogenetic tree was supported with bootstrap values based on 1000 replications.

**2.5 Fluorescence *in situ* hybridization (FISH) of *Alcanivorax* bacteria**
FISH probes for 16S rRNA gene of *Alcanivorax* bacteria were designed based on the 16S
rRNA gene sequence extracted from the *Alcanivorax* draft genome. Two 16S rRNA
fragments, 5'- CCTCTAATGGGCAGATTC-3' and 5'-CCCCCTCTAATGGGCAGA-3',
were selected as candidate probes with Probe_Design in the ARB package (Ludwig et al.,
2004). The coverage efficiency of the probes was then examined in the Silva database
(Quast et al., 2013). The 6-FAM-labeled probe used to target the *alkB* gene was
5'-ATGGAGCCTAGATAATGAAGT-3' (Wang et al., 2010). A pure culture of
*Alcanivorax borkumensis* Sk2 (Yakimov et al., 1998) was first used to examine the
probes before performing the assay, and a culture of *Escherichia coli* was used as a
negative control. Two grams of anhydrite crystals were sonicated for 30 min in 1 U DNase
I solution (Takara, Dalian, China). The crystals were washed with deionized water and then
ground into a powder with a beadbeater in a germ-free environment. The supernatant was
mixed with 37% formaldehyde (final concentration, 1-4%). To fix the cells in phosphate
buffer saline solution, the sample was maintained at 4℃ for 3-4 hr. After centrifugation at
13,000 r/min for 3 min, the supernatant was discarded. The remaining microbes were
soaked in 200 μL of PBS buffer, followed by addition of 200 μL of ethanol (Pernthaler et
al., 2002). The sample was filtered through 3-μm and 0.22-μm membranes sequentially
(diameter, 25 mm; Millipore, Eschborn, Germany). After dehydration of the membrane
using alcohol, 2 μL of dying solution containing oligonucleotide probes and 20 μL of
buffer (360 μL of 5 M NaCl,40 μL of 1 M Tris/HCl,700 μL of 100% formamide,2 μL
of 10% SDS,and water to a total volume of 2 mL). The hybridization of the probes to the
microbes was performed for 2 h at 46℃. Rinsing buffer (700 μL of 5 M NaCl, 1 mL of 1
M Tris/ HCl, 500 μL of 0.5 M EDTA, 50 μL of 10% SDS and water to a total volume of
50 mL) was used to remove free probes. For counterstaining, 50 μL of
4',6'-diamidino-2-phenylindole (DAPI) (Thermo Fisher, Carlsbad, USA) solution (1
μg/mL) was added to the sample. After incubation for 3 min, the sample was washed in
Milli-Q water (MetaPhor Bioproducts, Rockland, Maine) and 96% ethanol for 1 min
(Pernthaler et al., 2002). The microscopic observation was conducted using an Olympus
BX51 (Olympus, Tokyo, Japan).

## 253 3 Results

### 254 3.1 Physicochemical profile and cell counts

A thick anhydrite layer was present at the bottom of the sediment core based on
naked-eye observation of the color and grain size. The anhydrite layer at depths ranging
from 177-198 cm consisted of coarse, agglutinated crystals, which corresponded to the
high percentage of large grains (78 wt% larger than 63 μm) (Fig. 2). The XRD analysis
further confirmed that the crystals in this layer were anhydrite. In contrast, halite
comprised the evaporites at depths of 12 cm, 63 cm, 105 cm and 222 cm. For the samples
at different depths, the DOC concentration was measured, and the highest value was
recorded at 183 cm (80.9 mg $L^{-1}$), which was even higher than the surface layer at 12 cm
(Fig. 3). In the 12 cm layer, the cell density was $3.2 \times 10^5$ cells per $cm^3$, whereas in the
layers at 63 cm, 105 cm and 222 cm, it was reduced by 88%, 92% and 96%, respectively
(Fig. 3). The cell density was also calculated as the number of cells per gram of sediment.
The results revealed a value of $7.1\times10^5$ cells per gram at a depth of 12 cm, which declined
more than 70% in the deeper layers. Although the cell density in the 183 cm layer
($6.7\times10^4$ cells/cm$^3$) was markedly lower than that in the 12 cm layer, it was higher than
those in the 105 cm and 222 cm layers.

The sediment as a whole is a highly reductive environment, as indicated by the low
nitrate, low nitrite and extremely high ammonium concentrations (Fig. 3). To determine
the time of the anhydrite layers at 177-198 cm, an age estimate was performed for several
layers. The sediment ages were estimated based on the radioisotope $^{14}$C of *G. sacculifer*
assuming a linear increment from the top (Fig. 2). The results obtained for the layers
above and below the anhydrite layer indicated a narrow range of 750-770 years between
153 cm and 198 cm (Table 1).

**3.2 Draft genome of the dominant bacterial species in anhydrites**
About 1.8 Gbp Illumina raw sequencing data were obtained for the anhydrite sample and
3.1 Gbp data were obtained for the adjacent control layer. The size of the anhydrite and
control metagenomes was 59 and 84 Mbp, respectively, after assembly (Table S1). The
microbial communities differed remarkably according to the taxonomic assignment of the
16S/18S rRNA gene fragments in the two metagenomes (Fig. 4). At the genus level, only
*Ochrobactrum* and *Alkanindiges* were common inhabitants in both samples. *Alcanivorax*
and *Bacillus* were also dominant genera in the anhydrite and the control, respectively. At
the phylum level, excluding the Proteobacteria, the two metagenomes had distinctive
phyla. The anhydrite contained archaea that were represented by the methanogenic
*Methanoculleus* (Barret et al., 2012); and fungi that consisted of the Ascomycota. In
contrast, the control sediment contained mainly Firmicutes, Bacteroides, Actinobacteria,
and Deinococcus-Thermus. At last, an invertebrate species, *Prototritia* sp. belonging to
Arthropoda, was identified in the anhydrite.

**3.3 Genome binning of an *Alcanivorax borkumensis* genome**
The binned draft genome from the anhydrite metagenome was 3,069,971 bp and
comprised 77 contigs. A partial 16S rRNA gene sequence (805 bp) was extracted from
the draft genome. Because the sequence was almost identical to that of *A. borkumensis*
Sk2 (99.9%) (see also genomic alignment in Fig. S2), we considered the binned draft
genome to be from a strain of *A. borkumensis*. As shown in Figure 5, a phylogenetic tree
based on the 16S rRNA gene sequences of the genus *Alcanivorax* indicated that the strain
clustered with *A. borkumensis* Sk2, an exclusive and ubiquitous hydrocarbon-degrading
bacterium (Schneiker et al., 2006;Sabirova et al., 2011). The strain name of the *A.*
*borkumensis* in the sediment core was ABS183. It was the only microbial species that
could be reliably separated from the metagenome.

The genome of *A. borkumensis* ABS183, despite containing gaps, was slightly smaller
than that of *A. borkumensis* Sk2 (accession number NC_008260; 3,120,143 bp),
suggesting that the draft genome of *A. borkumensis* ABS183 was nearly complete. Also,
there were not detectable alignment gaps between the two genomes (Fig. S2). The
identification of a complete list of single-copy genes also supported the completeness of
the genome. The DDH estimation between *A. borkumensis* ABS183 and Sk2 was
97.1%±1.3%, which was higher than the standard cut-off value of 70% for genome
relatedness between pairs of species (Wayne et al., 1987). The ANI value between
ABS183 and Sk2 was 99.9%, which was also higher than the standard ANI criterion for
species identity (95%–96%) (Richter and Rossello-Mora, 2009). These results further
confirmed that ABS183 was a strain of *A. borkumensis*.

The genome of *A. borkumensis* ABS183 contains two copies of the alkane-1
monooxygenase gene (*alkA*; 10502_28 and 2890_35), which is an essential functional
gene for alkane utilization by *Alcanivorax* bacteria (Fig. 6) (Schneiker et al., 2006).
Neighboring the *alkA* genes, *alkBGHJ* genes, a GntR family transcriptional regulator
gene, and a rubredoxin gene were identified. The gene order of the related genes was
consistent with that of the homologs in the genome of strain Sk2 (Schneiker et al., 2006).
The *alk* genes were completely absent from the control metagenome. Moreover, the
genome of *A. borkumensis* ABS183 contains genes responsible for the reduction of
nitrogen oxides (KEGG genes: K00370-K00374 and K00362-K00363; nitrate reductase I
genes and nitrite reductase genes). The reduction process was believed to generate
ammonia for the efficient synthesis of amino acids (Schneiker et al., 2006). Ammonia
might be generated through nitrate reduction as indicated by the presence of the related
genes encoding nitrate and nitrite reduction enzymes (Fig. 6). Ammonia might be
imported by transmembrane transporters and assimilated into glutamate. A high demand
for fatty acids was a characteristic of *A. borkumensis* to perform rapid energy and organic
carbon storage. *A. borkumensis* ABS183 was probably able to synthesize long fatty acids
because the *fas* and *fabBFGIKZ* genes responsible for the elongation of fatty acids were
all present in its draft genome. In contrast, the essential *fas* gene (K11533) and other
relevant genes were not found in the control metagenome. Crude oil generally contains
aromatic compounds, and the current sediment at the sampling site also contained oil
(Wang et al., 2011). As expected, the two metagenomes possessed a complete set of
genes responsible for the degradation of aromatic compounds. Based on the homology of
the genes, the *Ochrobactrum* and *Alkanindiges* species probably played a role in this
degradation.

**3.4 Detection of bacteria in anhydrite crystals by DAPI and FISH**
To determine whether complete microbial cells could be maintained in the anhydrite
crystals, DAPI and FISH assays were conducted to visualize the microbes. The DAPI
results revealed the presence of complete cells that were released or embedded in the
crystals (Fig. 7A, D, and H). However, the FISH assay, which was used to detect *A.*
*borkumensis* ABS183 with two probes specific to the 16S rRNA gene, showed some
fluorescence-labeled microbes (Fig. 7B, E and I). These microbes could also be
envisioned with the FISH assay using the *alkB* gene probe (Fig. 7F and 7J). The *alkB* is
one of the functional genes that participate in alkane degradation (Schneiker et al., 2006).
The rod shape of the fluorescent microbes is consistent with the microscopic features
reported previously (Sabirova et al., 2011). These results indicated that some microbes in
the microscopic fields were *A. borkumensis* ABS183, as revealed in the anhydrite
metagenome.

**4 Discussion**

In the present study, we detected complete microbial cells and analyzed their metagenome in the anhydrite crystals from a deep-sea anoxic basin. The dominant bacterial species was *A. borkumensis* ABS183, an aerobic bacterium that is capable of degrading alkanes in crude oil. *Alcanivorax* is one of the bacterial indicators for the spilling of oil in waters and surface sediment (Yakimov et al., 2007). However, the Atlantis II brine pool is anaerobic and increasingly hydrothermal (Bougouffa et al., 2013b). The brine sediment in the basin was also found to be anoxic. Thus, *A. borkumensis* ABS183 could not be current inhabitants of the hydrothermal anoxic basin. This difference did not explain the stratification of microbial communities in the different sediment layers of the brine-filled basin. A recent study showed that *Alcanivorax* was not present in all sediment layers of a sediment core from the Atlantis II basin (Wang et al., 2015). A reasonable explanation for this finding is that the anhydrite layer at 177-198 cm in the sediment core was formed at a previous benthic site when hydrothermal solution was injected into the seafloor. The organisms living in the benthic water and subsurface sediment were subsequently sealed and protected in the anhydrite crystals. Because the metabolism of *A. borkumensis* bacteria was specifically used for the degradation of alkanes and other hydrocarbons in crude oil (Yakimov et al., 2007), the benthic site in which the anhydrite layer formed was probably an oil-spilling or oil-forming environment in the Atlantis II basin. The current hot sediments in the basin are biogenic and abiogenic sources of crude oil (Simoneit, 1988). Seeping of the oil has resulted in proliferation of *A. borkumensis* bacteria in the bottom water. Similarly, oil-utilizing bacteria were nourished after the oil-spilling disaster in the Gulf of Mexico (Gutierrez et al., 2013). The *A. borkumensis* bacteria were important producers of organic carbons as they could convert alkanes and nitrate into organic matter. Fatty acids and lipopolysaccharides that were yielded by *A. borkumensis* bacteria were nutrients for the whole ecosystem.

Based on the results in the present study, we propose that mild eruptions of hydrothermal solutions injected calcium-rich solutions into the seafloor and produced anhydrite veins by mixing with sulfate in the bottom water of the Atlantis II rift basin. The anhydrite layer was then covered by sulfide minerals and biological debris such as the planktonic foraminifera *G. sacculifer*. In this study, we narrowed the age of the thick anhydrite layer

to 750-770 years using $^{14}$C isotope of the *G. sacculifer* specimens. This result also
indicates a relatively young sediment age and a high accumulation rate of precipitated
metals in the Atlantis II basin. Because the upward movement of hydrothermal solutions
might transfer some foraminifera specimens from lower layers to the anhydrite layer, we
did not use the foraminifera between anhydrite crystals. In our previous study, we have
shown evidence of oil formation in the Atlantis II brine pool (Wang et al., 2011). The
organic carbon content can be converted to aromatic compounds under the hydrothermal
conditions in the pool based on chemical and metagenomic evidence (Wang et al., 2011).
However, the bottom of the anoxic brine pool was not a habitat of *Alcanivorax* species
(Bougouffa et al., 2013a;Blanc and Anschutz, 1995), suggesting that *Alcanivorax*
flourished in the basin before the formation of brine water layers over the sediment
(Blanc and Anschutz, 1995).

Although there were differences in microbial communities between the anhydrite crystals
and the control sediment, *Ochrobactrum* sp. was one of the common inhabitants.
Previous studies have shown that *Ochrobactrum* species could metabolize aromatic
compounds aerobically and anaerobically (Zu et al., 2014;Mahmood et al., 2009), which
explains their presence in both metagenomes assessed in the current study. Moreover, we
determined the concentrations of nitrogen oxides in the different sulfide layers, although
only low concentrations were detected. *Ochrobactrum* species were potentially able to
anaerobically degrade polycyclic aromatic compounds using nitrate as an oxygen donor
(Mahmood et al., 2009;Wu et al., 2009). Such a chemolithoheterotrophic lifestyle is in
accordance with the current *in situ* environment of the sediment in the Atlantis II.
Regardless of the environmental changes indicated by the findings in the present study,
the spreading of *Ochrobactrum* sp. was seemingly not affected. Although the
metagenomes in the present study contained an abundant essential genes for degrading a
variety of aromatic compounds, the microbial degradation of these compounds might
have been attenuated by a lack of oxygen and a high level of salinity (Klinkhammer and
Lambert, 1989). Anaerobic degradation of compounds is more difficult than aerobic
degradation, often requiring oxygen donors such as nitrate and sulfate (Mahmood et al.,
2009;Wu et al., 2009). Based on its ability to survive under anoxic conditions,
*Ochrobactrum* sp. is probably able to maintain a higher level of fitness in the control
sediment compared with *Alcanivorax*. In the present study, the *Alkanindiges* identified in
both metagenomes was also a well-known alkane degrader (Klein et al., 2007;Bogan et
al., 2003). Because of its presence in both anhydrites and the adjacent sulfide layer, we
assumed that the *Alkanindiges* bacterium was also capable of surviving aerobically and
anaerobically in the oil-producing sediment. Hence, the change from an oxic to an anoxic
benthic environment caused a dramatic shift in the microbial communities, resulting in
the extinction of the obligate aerobic alkane-utilizer *Alcanivorax* and continuous
residency of anaerobic oil-degraders. The availability of nitrogen oxides and the
dissolution of sulfate from anhydrite crystals were possibly critical to the metabolic
activities of the anaerobes. In addition, the *Bacillus* and fungi present in the control
sediment were probably present in the form of dormant spores. In a recent report,
*Ochrobactrum* and *Bacillus* were confirmed to be dominant species in some upper sulfide
layers in the Atlantis II (Wang et al., 2015). Altogether, in the present study, the current
microbial inhabitants in the sulfide layers were largely different from those in the
anhydrite crystals.

The geochemical data collected herein suggested that the sub-superficial anhydrite layer
could release organic carbon contents into the sediment, as reported previously [12,13]. Our
measurement of DOC at 80.9 mg $L^{-1}$ in the anhydrite layer was higher than the generally
accepted maximum value of 50 mg $L^{-1}$ for marine sediments (Cameron et al., 2006). The
abnormally high DOC was considered a notable alteration of the local environments,
probably resulted from the breakdown of anhydrite crystals. Anhydrites in the Atlantis II
brine sediment were likely maintained by the high salinity and temperature, and then
slowly dissolved. This phenomenon may be explained by the slight undersatuation of the
anhydrite in the Atlantis II sediment (Anschutz et al., 2000). Such anhydrite layers are
widely distributed in Middle Eastern sediments (Alsharhan and Nairn, 1997). Hence, our
findings shed light on the formation of micro-environments by anhydrite evaporites in the
deep sediments. In this study, there was an inconsistency between the cell density and the
DOC at the 12-cm depth layer, in which the DOC could not support a 10-fold higher
biomass. This phenomenon probably resulted from the formation of petroleum
compounds under the hydrothermal effects (Wang et al., 2011). In the petroleum,
hydrophobic organic compounds (HOCs) consisting of polycyclic aromatic hydrocarbons
(PAHs) could not be counted in our DOC measurements (personal communication with J.
Pearsons). The nutrient supply is critical for microbes to survive in deep-sea sediment.
Apart from the chemolithoautotrophic microbes, numerous other inhabitants take
advantage of the buried organic matter. Importantly, the trapped organic matter serves as
a nutrient supply following the dissolution of organic-rich anhydrite crystals. Therefore,
our findings highlighted the importance of the nutrients released from the anhydrite facies
for microbes in deep-sea subsuperficial sediment.

*Author contributions.* Y. Wang, T.G. Li, and P. Y. Qian were responsible for the study
design. Data analysis was performed by Y. Wang, T.G. Li, J. T. Li, Q. L. Lai, and Z. M.
Gao. M.Y. Wang conducted FISH assay. The manuscript was prepared by Y. Wang with
contributions from all co-authors.
*Acknowledgments.* This study was supported by the National Science Foundation of
China No. 41476104 and No. 31460001 to Y. Wang, and the King Abdullah University
of Science and Technology (SA-C0040/UK-C0016) to P.Y. Qian. This work was also
supported by Hainan international collaborative grant No. KJHZ2015-22.

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

**Data Accessibility**
Illumina raw data will be accessible under SRA356974 in the NCBI SRA database. *A.*
*borkumensis* ABS183 genome was deposited at the NCBI database: BioProject
LKAP00000000.












**Table 1.** Age estimates of the sediment layers

| Layer (cm) | Age (year) | Age error (year) |
|---|---|---|
| 3-6 | 320 | 25 |
| 21-24 | 475 | 35 |
| 45-48 | 490 | 30 |
| 90-93 | 500 | 25 |
| 129-132 | 560 | 35 |
| 153-156 | 750 | 30 |
| 198-201 | 770 | 30 |
| 222-225 | 880 | 30 |


Eight sediment layers were selected for the age estimates using radioisotope $^{14}$C of *G.*
*sacculifer* collected from the respective layers. The age was corrected by the 400-year
reservoir age with an error range.


**Figures**
**Figure 1. Anhydrite crystals and genome binning.**
Anhydrite crystals in a Petri dish (90 mm in diameter) (A) were used for DNA extraction.
The amplified genomic DNA was sequenced and then reassembled. Based on the G+C
content and read coverage, the binned contigs with high coverage levels (B) were
selected for examination of the tetranucleotide frequency consistency in the PCA analysis
(C).

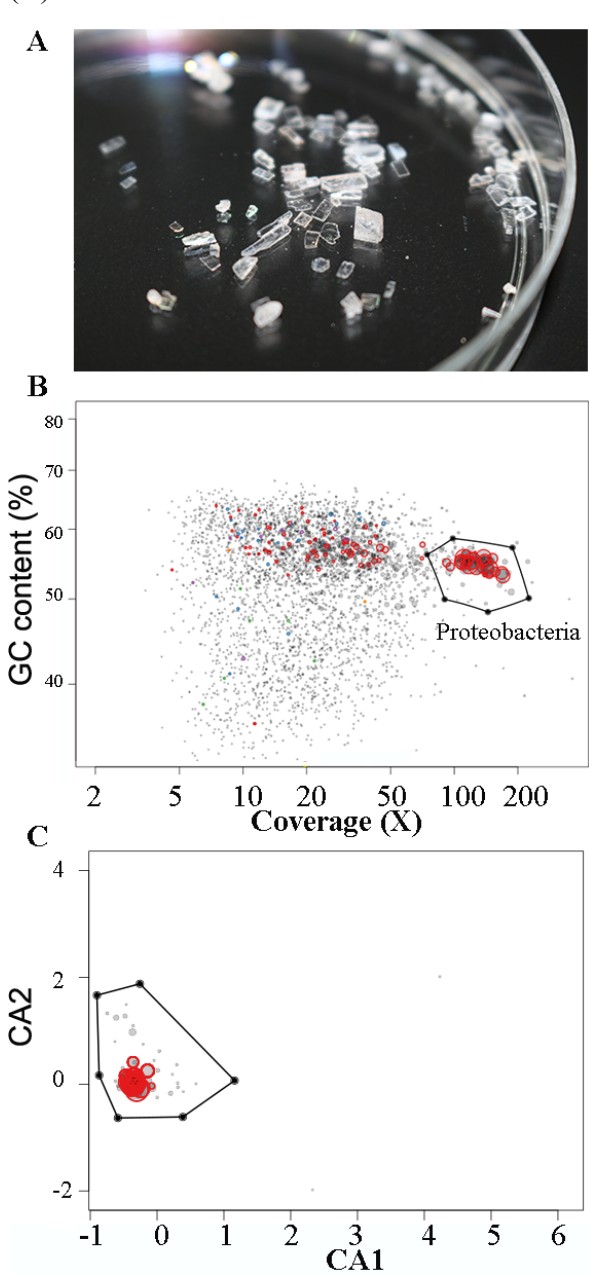



**Figure 2. Grain size and age of selected layers**
The percentages of the small particles (<63μm) in dry weight are shown for 75 slices of
the sediment core (small squares on the line). The age estimates (black circles) of the
selected layers were performed using radioisotope 14C of the
monospecific *Globigerinoides sacculifer* specimens. Age errors ranged between 25 to 40
years. Anhydrite and control layers for metagenomic study were indicated by arrows.

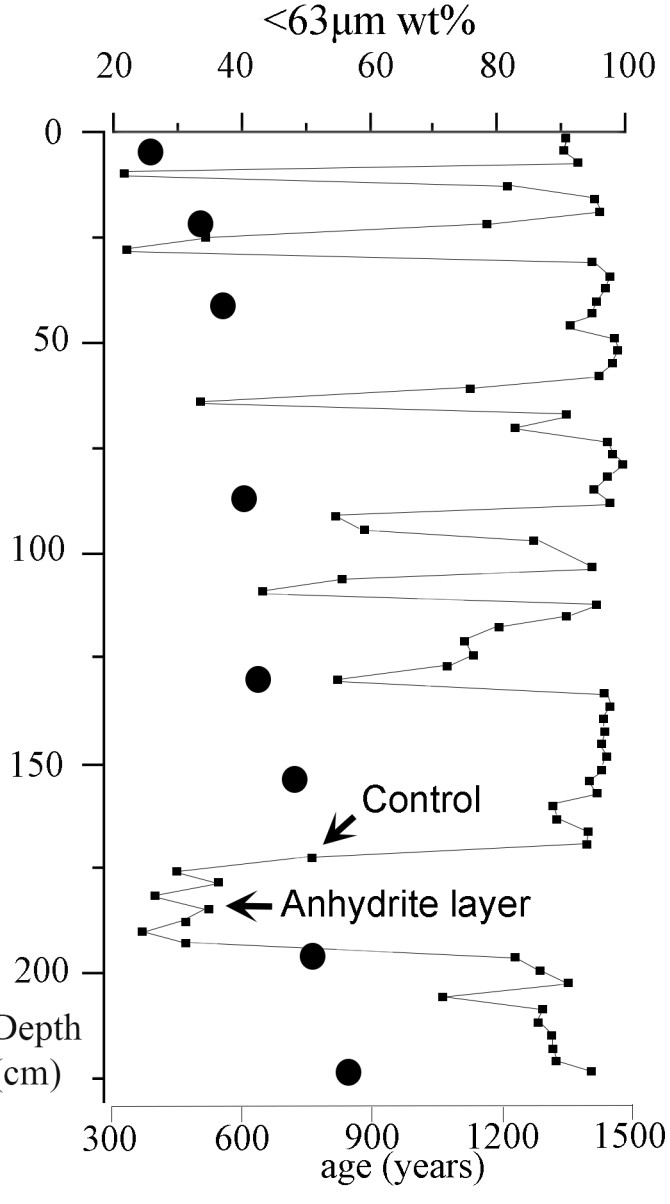




**Figure 3. Nutrient measurements and cell counts in the different sediment layers.**
The pore water samples were analyzed for five layers of a sediment core obtained from
the Atlantis II Deep (21°20.76' N, 38°04.68' E) in 2008. DOC: dissolved organic carbon.

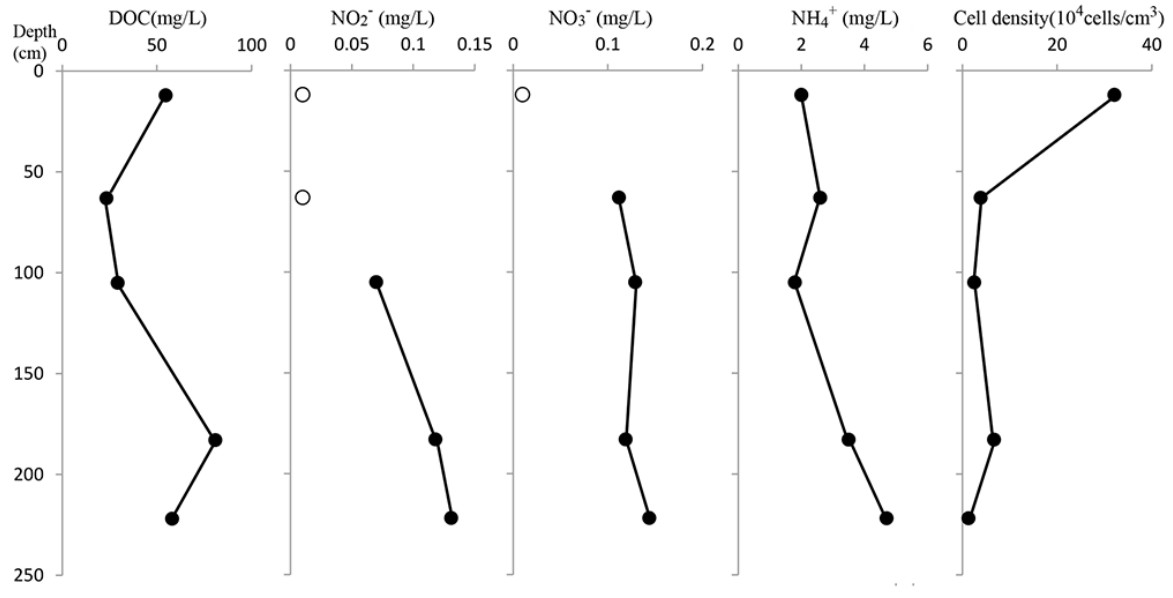

758    759    760    761    762    763    764    765    766    767    768    769    770    771    772    773    774    775

**Figure 4. Microbial communities in anhydrite crystals and neighboring control**
**sediment.**
Phyla and genera in the anhydrite crystals and control layer were predicted using 16S/18S
rRNA gene fragments extracted from the corresponding metagenomes (D-T:
Deinococcus-Thermus). The relative abundance of the genera can be estimated by the
coverage level (>5) of the 16S/18S rRNA fragments by reads.

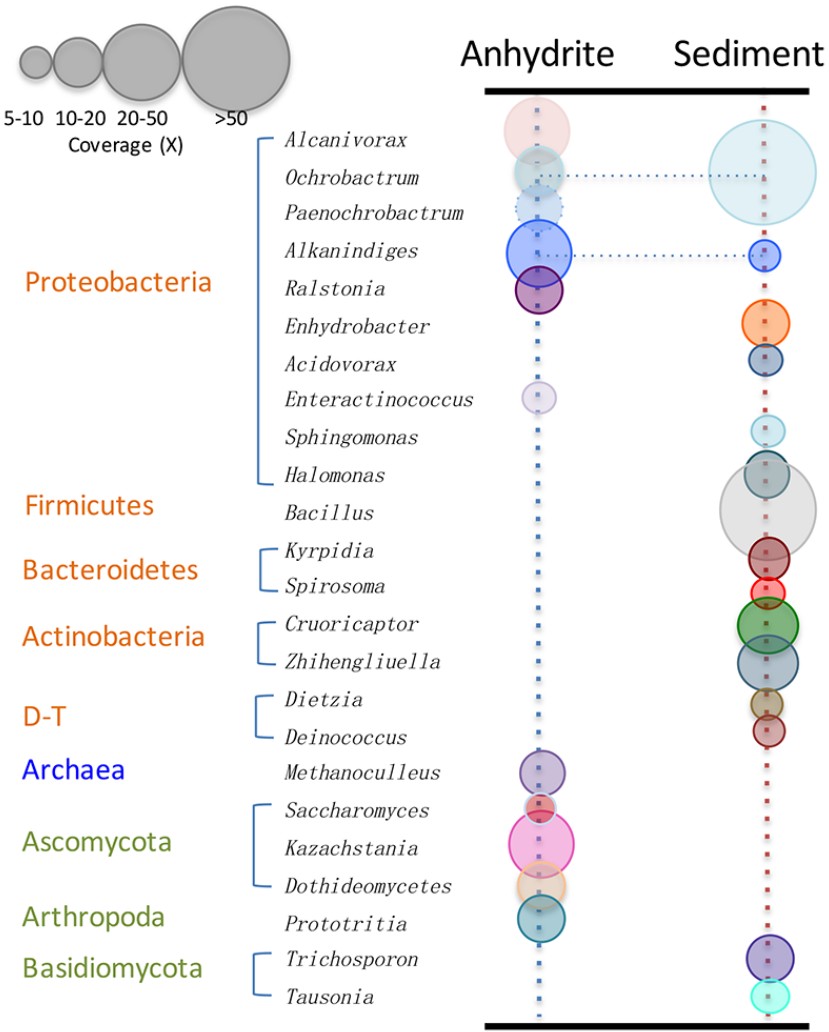










**Figure 5. Phylogenetic tree of 16S rRNA genes.**

Bootstrap values (expressed as percentages of 1000 replications) are shown at the

branches of the neighbor-joining tree.

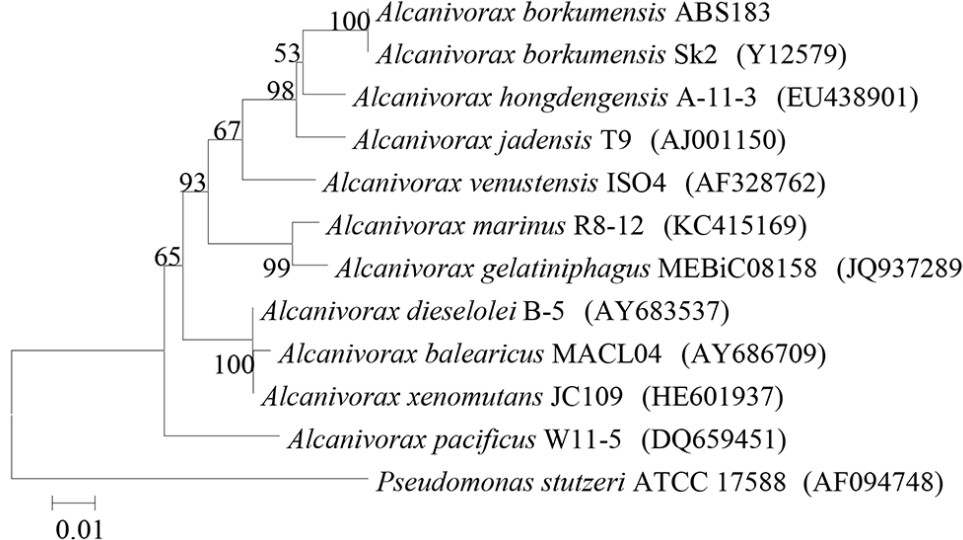

794     0.01


**Figure 6. Schematic model of metabolism and cross-membrane transporters**

The model was predicted based on the genes in the draft genome of *Alcanivorax*

*borkumensis* ABS183.

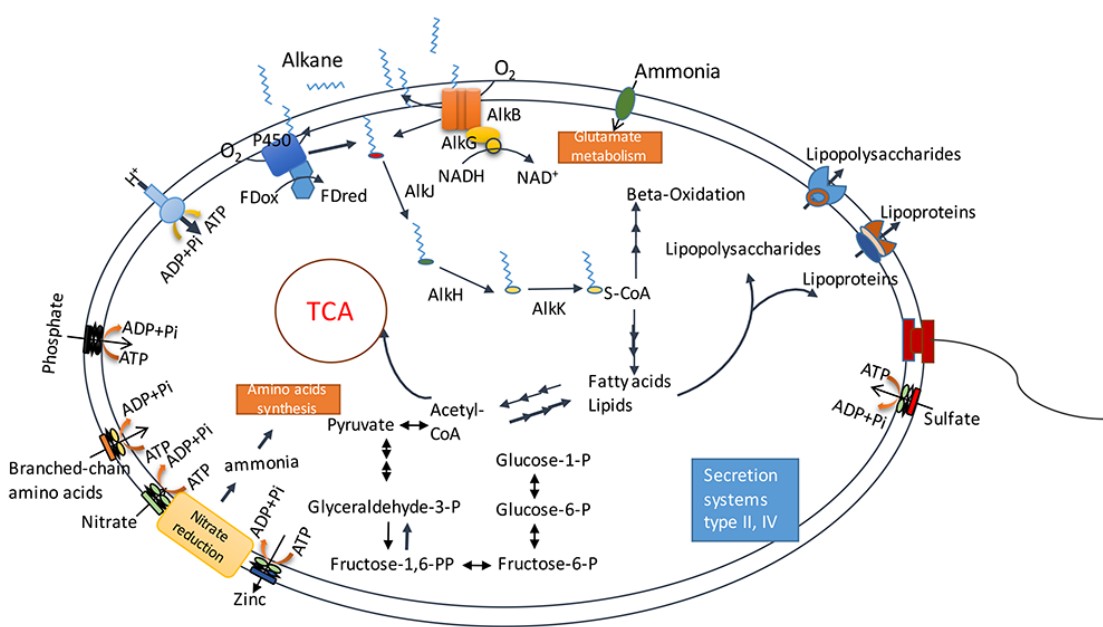





**Figure 7. Fluorescence *in situ* hybridization (FISH) of *Alcanivorax* sp. ABS183 embedded in anhydrite crystals.**

DAPI staining and FISH using 16S rRNA probes are shown in A and B. The merged image of A and B is shown in Fig. 7C. DAPI staining and FISH were also performed using two samples that were filtered with 3-μm (D-G) and 0.22-μm (H-K) membranes, respectively. *Alcanivorax* bacteria were released from the large crystals filtered through the 3-μm membranes (D-G). The bacteria were stained with DAPI (D), 16S rRNA probes (E) and the *alkB* probe (F), respectively, and overlaid (G). Using a sample filtered through a 0.22-μm membrane, a dividing *Alcanivorax* sp. ABS183 cell was labeled using the same method and probes (H-J). The microscopic fields shown in H-J are merged in Fig. 7K.

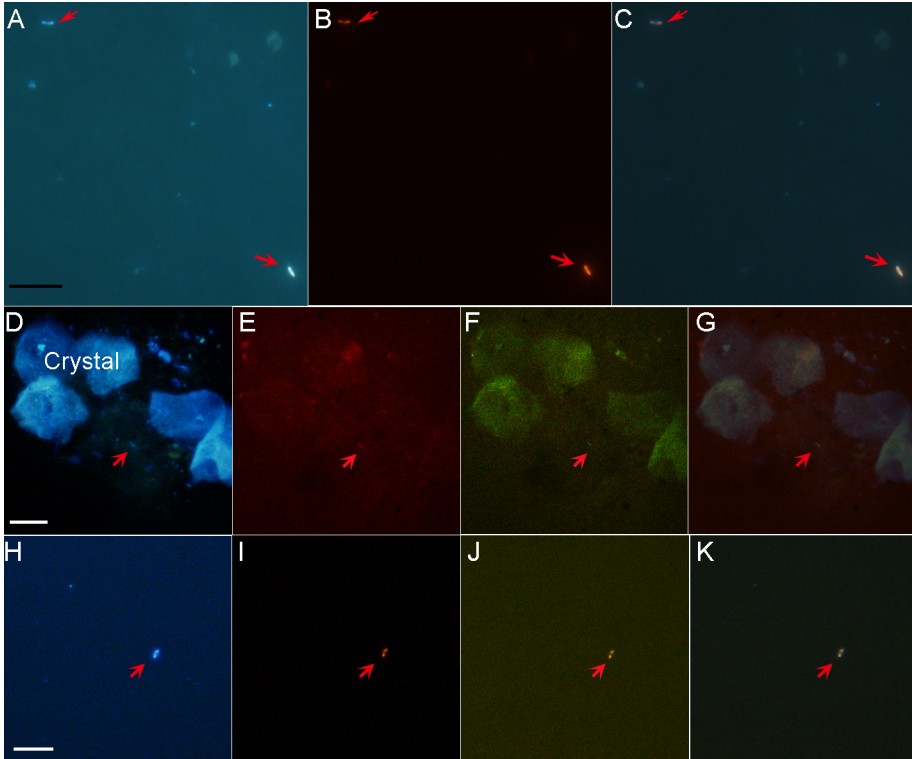

813

814

815