# Peer review of "Archive of bacterial community in anhydrite crystals from a"

_Biogeosciences, 2016_

## Referee Comment (RC1) · M. Bomberg (Referee) · 19 Sep 2016

The paper by Wang et al describes the microbial community associated with anhydrite crystals in a deep sea sediment basin located at more than 2 km depth in the Red Sea. According to the authors the microbial communities detected could verify past oil-spilling events to a relatively good accuracy. The microbial communities were investigated using metagenomic tools and the authors found that alkane-degrading Alcanivorax species were dominant in the metagenomes coinciding with the past oil-spill. Several metabolic alkane-degradation pathways were detected. The microbial community of the anhydrite crystals were investigated using fluorescent in situ hybridization for identification of Alcanivorax cells that contained alkB genes. The chemistry and biological parameters of the sediment core from which the anhydrite crystals originated, differed with depth. A clear peak in organic carbon content and a significant peak at specific depth corresponding to the oil-spill. The authors conclude that the organic carbon stored in the sediments, e.g. as anhydrite crystals, is slowly released for the benefit of the whole sediment microbial community. The text as a whole is quite clear and the subject is interesting. The materials and methods could benefit from more information. I would also like to see some mote of the genome of the Alcanvorax and some metabolic pathway maps. The fact that a new uncultured Alcanivorax has been detected could be discussed more as well as its role in carbon cycling in deep sediments.

Specific comments; L115, how did you get the supernatant? Did you let the solids sedimentate first or did you use centrifugation? L175, give more information about what you did with R and which packages you used. L178, what is HMM L203, what label did you use? Did you have a nonsense probe to check for unspecific labeling? L223, MQ water? L234-251, check the figures. I thing the reference to Fig 2 should be Fig 3 and vice versa. L255 ->, did you get any sequences from the blanks and if so, what did you do about it? Results&Discussion, I would like to see some figures with the Alcanivorax genome and relevant metabolic maps. How does the Alcanivorax interact with the rest of the microbial community?

Typos; L58, change 'were' to 'are' L81, do you mean 'found', not 'located'? L137, correct 'grounded' to 'ground' or 'homogenized'

---

## Referee Comment (RC2) · Anonymous Referee #2 · 20 Sep 2016

The manuscript submitted by Wang et al. describes the diversity of organisms in anhydrite crystals and further states a high abundance of Alcanivorax microbes. The authors see the specific community composition of the 750 years old crystals as indicator of an oxic oil-spilling benthic environment. Overall, the approach and results are highly interesting and definitely worth reporting in Biogeosciences.

I, however, suggest revising and clarifying several parts of the manuscript before final publication.

General remarks:

While I do not doubt the results of the metagenomic investigation per se, I am honestly wondering what the meaning of the quantification of microbes based on traces of DNA is and how you could find intact cells being preserved for 750 years in crystals, which are still traceable using FISH. I am missing a detailed explanation and discussion on that topic as also for me the microscopic pictures are not entirely satisfying. Additionally I would like to see some proof of the assembled genome and a metabolic map. Further, the conclusion on past oil-spilling is a bit too loosely connected to the other parts of the story, both in the abstract and the results and discussion parts. This could be done by introducing geochemical data earlier and by rephrasing some of the respective text parts. The references are not in Copernicus style.

Specific comments:

Abstract:

First sentence should be rephrased as a hypothesis.

l.33: which binning tool? Replace separated by assembled

l. 35. Remove 'the'

l.33-l.37: you are jumping back and forward between the metagenome and the cell identification

l. 40 sentence is misplaced, here. Should probably be moved upwards.

l. 41-45: As written this seems to be highly speculative

Introduction

L. 54: Repetition of previous sentence.

l. 57: I don't understand this sentence.

l. 59: replace 'prediction' by ' interpretation'

l. 60: replace 'have been' by 'are'

l. 63: The previous sentence already started with 'although'- replace.

l. 64: I would like to see a reference for this.

l. 69: replace 'become' by ' may get' and add a reference for this statement.

l. 79: the bodies are the biomarkers? Actually lipids, DNA or also pigments may be used as biomarkers. Not sure whether the word 'body' is really the right expression for dead microbial cells.

l. 81: similar to what?

l. 95: 'Hence' is contradictory to 'probably'

l. 97: Which other markers? Why probably? This would actually be the place to explain the geochemical background.

l. 100-104. This belongs to the methods part

Material and methods

l. 110: A map of the sampling location would be beneficial.

l. 138: Commonly, all abbreviations should be introduced when used for the first time.

l. 146: this is a very low amount of DNA, how representative is this for community analysis? In how far is the quantification of OTUs and the genome identification trustworthy based on such low amounts of DNA? What about different degradation patterns of different organisms?

l. 150: Is this a random amplification?

l. 156: Which chemistry and protocol?

Some statistics on the sequencing (how many reads per run, how many of the identifiable, how many reads related to 16S rDNA, etc.) would be desirable.

l. 169: How?

l. 169: Based on the low amount of DNA and amplification steps in between, I doubt the quantitative aspects of the analysis.

l.175: remove gap after ). Rstudio will not make any visualization without a prober script, so please provide details on your script. Also, provide the correct reference instead of the link to the homepage.

l. 181: It is unclear how the draft genome was assembled. What is the completeness? A genome plot would increase the credibility.

l. 207: replace 'examine' by 'validate'

l. 223 MQ- please spell out

I doubt that FISH produces a reliable signal if you only get 20pg of DNA out of the crystals.

Results

l.232, 236, 238, 246, 249 : Check numbering of figures. Also, where do you refer to Fig. 5?

l. 262: archaea, fungi without capital letters

l. 265: Why is that surprising?

l. 269 onwards: this needs a genome plot and a Kegg metabolic map.

l. 278: I don't buy the quantitative aspect, here.

l. 346: postulate seems to be a bit strong for the line of evidence provided, here.

l. 346 ff: This information is what you need to put into the abstract to make it convincing

---

## Author Comment (AC1) · 29 Sep 2016

Reviewer 1: The paper by Wang et al describes the microbial community associated with anhydrite crystals in a deep sea sediment basin located at more than 2 km depth in the Red Sea. According to the authors the microbial communities detected could verify past oil-spilling events to a relatively good accuracy. The microbial communities were investigated using metagenomic tools and the authors found that alkane-degrading Alcanivorax species were dominant in the metagenomes coinciding with the past oil-spill. Several metabolic alkane-degradation pathways were detected. The microbial community of the anhydrite crystals were investigated using fluorescent in situ hybridization for identification of Alcanivorax cells that contained alkB genes. The chemistry and biological parameters of the sediment core from which the anhydrite crystals originated, differed with depth. A clear peak in organic carbon content and a significant peak at specific depth corresponding to the oil-spill. The authors conclude that the organic carbon stored in the sediments, e.g. as anhydrite crystals, is slowly released for the benefit of the whole sediment microbial community. The text as a whole is quite clear and the subject is interesting. The materials and methods could benefit from more information. I would also like to see some mote of the genome of the Alcanvorax and some metabolic pathway maps. The fact that a new uncultured Alcanivorax has been detected could be discussed more as well as its role in carbon cycling in deep sediments. Response: Thanks for the comments. We have put more details in the MM. In the Results, we made a schematic map for the Alkanivorax bacteria. Regarding the role of Alcanivorax, we put several lines in Discussion, lines 451-453.

Specific comments; L115, how did you get the supernatant? Did you let the solids sedimentate first or did you use centrifugation? Response: we put the sample for 30 min and did not centrifuge. We modified the corresponding place. L175, give more information about what you did with R and which packages you used. Response: yes, we inserted the linkage of the R scripts. L178, what is HMM Response: HMM means hidden markov model. We have deleted it. L203, what label did you use? Did you have a nonsense probe to check for unspecific labeling? Response: actually we used E. coli to examine the probe as a negative control. We have inserted this in the new version. L223, MQ water? Response: we have spelled out. L234-251, check the figures. I thing the reference to Fig 2 should be Fig 3 and vice versa. Response: yes, modified. L255 ->, did you get any sequences from the blanks and if so, Response: The sequences in blanks were dimers of the primers for the amplification, which can be justified from the size of the smear. There were not any long amplified sequences from contaminated DNA. ResultsDiscussion, I would like to see some figures with the Alcanivorax genome and relevant metabolic maps. How does the Alcanivorax interact

with the rest of the microbial community? Response: yes, we inserted a metabolic map for the Alcanivorax bacterium. Regarding the interaction, we inserted lines in Discussion, (lines 451-454). They are supposed to a convertor between alkanes and organic carbons and nitrogen. Typos; L58, change 'were' to 'are' L81, do you mean 'found', not 'located'? L137, correct 'grounded' to 'ground' or 'homogenized' Response: yes, all are corrected.

Please also note the supplement to this comment:
http://www.biogeosciences-discuss.net/bg-2016-204/bg-2016-204-AC1-
supplement.pdf
* * *
[Figure]

[Figure]

**Fig. 1.**

[Figure]

**Fig. 2.**

**Supplement:**

Figure S1 Geographic map of the sampling site.

[Figure]

The gravity core was obtained from the bottom of the Atlantis II Deep (ABP) in 2008.

Figure S2. Genomic comparison between *Alcanivorax borkumensis* Sk2 and ABS183.

[Figure]

The comparison was conducted in ACT website (webact.org).

---

## Author Comment (AC2) · 29 Sep 2016

Reviewer 2 The manuscript submitted by Wang et al. describes the diversity of organisms in anhydrite crystals and further states a high abundance of Alcanivorax microbes. The authors see the specific community composition of the 750 years old crystals as indicator of an oxic oil-spilling benthic environment. Overall, the approach and results are highly interesting and definitely worth reporting in Biogeosciences. I, however, suggest revising and clarifying several parts of the manuscript before final publication. General remarks: While I do not doubt the results of the metagenomic investigation per se, I am

honestly wondering what the meaning of the quantification of microbes based on traces of DNA is and how you could find intact cells being preserved for 750 years in crystals, which are still traceable using FISH. Response: Since 2009, we have been working on this sediment layers for several years to answer the question regarding the elevated organic carbons in the ABS183 layer. The hypothesis is the preservation of cells in the crystals as we also revealed eukaryotic and archaeal sequences from metagenome. It was very surprising for us to find intact microbes in the crystals. I am missing a detailed explanation and discussion on that topic as also for me the microscopic pictures are not entirely satisfying. Response: we have to apologize for the quality of the pics. We selected these pictures to illustrate the presence of the microbes from hundreds of pictures. Although the quality is still not satisfying for you, the result is almost our best for readers. As we can understand, these microbes had been preserved for hundreds of years. It's difficult to find out well-preserved and well-labeled microbes. Additionally I would like to see some proof of the assembled genome and a metabolic map. Further, the conclusion on past oil-spilling is a bit too loosely connected to the other parts of the story, both in the abstract and the results and discussion parts. This could be done by introducing geochemical data earlier and by rephrasing some of the respective text parts. The references are not in Copernicus style. Response: thanks for these comments. We have finished a new schematic map for the metabolism based on the gene profile. About the context of oil-spilling, we have rephrased the Abstract and inserted several lines in the Introduction (lines 139-157) to introduce the geochemical background in the oil-producing hydrothermal sediment. Specific comments: Abstract: First sentence should be rephrased as a hypothesis. Response: yes l.33: which binning tool? Replace separated by assembled Response: we assembled the reads and then conducted binning of the contigs for a draft genome. l. 35. Remove 'the' l.33-l.37: you are jumping back and forward between the metagenome and the cell identification Response: yes, these sentences are reorganized. l. 40 sentence is misplaced, here. Should probably be moved upwards. Response: yes l. 41-45: As written this seems to be highly speculative Response: yes, we have changed the tone. Introduction L. 54:

Repetition of previous sentence. Response: yes , deleted. l. 57: I don't understand this sentence. Response: rephrased! l. 59: replace 'prediction' by ' interpretation' Response: yes l. 60: replace 'have been' by 'are' Response: yes l. 63: The previous sentence already started with 'although'- replace. Response: yes, used 'but' now. l. 64: I would like to see a reference for this. Response: yes, two refs inserted. l. 69: replace 'become' by ' may get' and add a reference for this statement. Response: yes. l. 79: the bodies are the biomarkers? Actually lipids, DNA or also pigments may be used as biomarkers. Not sure whether the word 'body' is really the right expression dead microbial cells. Response: yes, it was replaced with 'evidence'. Line 118 l. 81: similar to what? Response: changed. Line 120 l. 95: 'Hence' is contradictory to 'probably' Response: yes. l. 97: Which other markers? Why probably? This would actually be the place to explain the geochemical background. Response: yes, we inserted several lines for the introduction. l. 100-104. This belongs to the methods part Response: this part was simplified. Material and methods l. 110: A map of the sampling location would be beneficial. Response: we used a suppl figure S1. l. 138: Commonly, all abbreviations should be introduced when used for the first time. Response: yes. l. 146: this is a very low amount of DNA, how representative is this for community analysis? In how far is the quantification of OTUs and the genome identification trustworthy based on such low amounts of DNA? What about different degradation patterns of different organisms? Response: We did not use up all the extraction. In total, there was not 1ng of DNA, so we have to use the amplification. The DNA might be used for many amplifications actually. The kit may allow for unbiased amplification of all sorts of DNA. We have a publication to support the linear amplification of microbial genomic DNA. l. 150: Is this a random amplification? Response: the theoretical basis and our test support the random amplification for the trace DNA. l. 156: Which chemistry and protocol? Some statistics on the sequencing (how many reads per run, how many of the identifiable, how many reads related to 16S rDNA, etc.) would be desirable. Response: we should have put all the information. The sequencing was done in a service centre. We don't know how many sequences were produced in one run. Perhaps there were

many samples in one runs. l. 169: How? Response: we inserted the command for this work (line2 239-240). l. 169: Based on the low amount of DNA and amplification steps in between, I doubt the quantitative aspects of the analysis. Response: we admit that this step is a brief statistics of the community. The abundance of the species could not be quantified in an accurate manner. l.175: remove gap after ). Rstudio will not make any visualization without a prober script, so please provide details on your script. Also, provide the correct reference instead of the link to the homepage. Response: yes. We inserted a linkage for the R scripts. See lines 247-248. l. 181: It is unclear how the draft genome was assembled. What is the completeness? A genome plot would increase the credibility. Response: the assembly work was introduced at line 236. The assembly made the reads into long contigs. Then the binning process grouped the contigs according to their distinctive coverage levels and tetranucleotide frequencies. The completeness was assessed firstly by the number of single-copy genes. We have used Fig. S2 to illustrate the genomic alignment between Sk2 and ABS183. From the alignment, there are no notable gaps between the two genomes, indicating the high genomic completeness of the ABS183 strain. l. 207: replace 'examine' by 'validate' Response: yes l. 223 MQ- please spell Response: yes I doubt that FISH produces a reliable signal if you only get 20pg of DNA out of the crystals. Response: actually we obtained far more than 20pg of DNA. We concentrated the cells into a membrane, so that the microbes from many crystals were released and condensed on one membrane. Results l.232, 236, 238, 246, 249 : Check numbering of figures. Also, where do you refer to Fig. 5? Response: yes, there are mistakes in the figures. The figure 5 was described in line 364. l. 262: archaea, fungi without capital letters. Response: yes. l. 265: Why is that surprising? Response: deleted. l. 269 onwards: this needs a genome plot and a Kegg metabolic map. Response: we described the metabolic map of Figure 6. Lines 392-403. l. 278: I don't buy the quantitative aspect, here. Response: we deleted it. l. 346: postulate seems to be a bit strong for the line of evidence provided, here. Response: we used 'proposed' now. l. 346 ff: This information is what you need to put into the abstract to make it convincing Response: this is a nice comment. See

the second sentence in the Abstract.

---

## Author Comment (AC3) · 29 Sep 2016

no more comments

[Figure]

**Archive of bacterial community in anhydrite crystals from a**
**deep-sea basin provides evidence of past oil-spilling in a**
**benthic environment in the Red Sea**

Yong Wang[1,2], Tie Gang Li[3,4], Meng Ying Wang[1], Qi Liang Lai[5], Jiang Tao Li[6], Zhao
Ming Gao[1], Zong Ze Shao[5], Pei-Yuan Qian[2,*]
[1]Institute of Deep-Sea Science and Engineering, Chinese Academy of Sciences, San Ya,
China
[2] Division of Life Science, Hong Kong University of Science and Technology, Clear
Water Bay, Hong Kong, China
[3] Key Laboratory of Marine Sedimentology and Environmental Geology, First Institute of
Oceanography, State of Oceanic Administration (SOA), Qingdao, China
[4]Laboratory for Marine Geology, Qingdao National Laboratory for Marine Science and
Technology, Qingdao, China
[5] Key Laboratory of Marine Biogenetic Resources, The Third Institute of Oceanography,
SOA, Xiamen, China
[6] State Key Laboratory of Marine Geology, Tongji University, Shanghai, China
**\*Corresponding author:** boqianpy@ust.hk, Tel: 852-2358-7331

**Keywords:** *Alcanivorax*; metagenome; anhydrite; Atlantis II brine pool; hydrothermal
sediment

**Running title:** Archive of microbial inhabitants in anhydrites

**Fig. 1.** maindoc

[Figure]

**Fig. 2.** maindoc with modifications

**Archive of bacterial community in anhydrite crystals from a**
**deep-sea basin provides evidence of past oil-spilling in a**
**benthic environment in the Red Sea**
Yong Wang[1,2], Tie Gang Li[3,4], Meng Ying Wang[1], Qi Liang Lai[5], Jiang Tao Li[6], Zhao
Ming Gao[1], Zong Ze Shao[5], Pei Yuan Qian[2,*]

Wang Yong 2016/9/28 8:59 PM

[1]Institute of Deep-Sea Science and Engineering, Chinese Academy of Sciences, San Ya,
China
[2] Division of Life Science, Hong Kong University of Science and Technology, Clear
Water Bay, Hong Kong, China
[3] Key Laboratory of Marine Sedimentology and Environmental Geology, First Institute of
Oceanography, State of Oceanic Administration (SOA), Qingdao, China
[4]Laboratory for Marine Geology, Qingdao National Laboratory for Marine Science and
Technology, Qingdao, China
[5] Key Laboratory of Marine Biogenetic Resources, The Third Institute of Oceanography,
SOA, Xiamen, China
[6] State Key Laboratory of Marine Geology, Tongji University, Shanghai, China
**\*Corresponding author:** boqianpy@ust.hk, Tel: 852-2358-7331
**Keywords:** *Alcanivorax*; metagenome; anhydrite; Atlantis II brine pool; hydrothermal
sediment
**Running title:** Archive of microbial inhabitants in anhydrites

---

## Author Response (AR1)

**Reviewer 1:**

The paper by Wang et al describes the microbial community associated with anhydrite crystals in a deep sea sediment basin located at more than 2 km depth in the Red Sea. According to the authors the microbial communities detected could verify past oil-spilling events to a relatively good accuracy. The microbial communities were investigated using metagenomic tools and the authors found that alkanedegrading Alcanivorax species were dominant in the metagenomes coinciding with the past oil-spill. Several metabolic alkane-degradation pathways were detected. The microbial community of the anhydrite crystals were investigated using fluorescent in situ hybridization for identification of Alcanivorax cells that contained alkB genes. The chemistry and biological parameters of the sediment core from which the anhydrite crystals originated, differed with depth. A clear peak in organic carbon content and a significant peak at specific depth corresponding to the oil-spill. The authors conclude that the organic carbon stored in the sediments, e.g. as anhydrite crystals, is slowly released for the benefit of the whole sediment microbial community. The text as a whole is quite clear and the subject is interesting. The materials and methods could benefit from more information. I would also like to see some mote of the genome of the Alcanvorax and some metabolic pathway maps. The fact that a new uncultured Alcanivorax has been detected could be discussed more as well as its role in carbon cycling in deep sediments.

Response: Thanks for the comments. We have put more details in the M&M. In the Results, we made a schematic map for the Alkanivorax bacteria. Regarding the role of Alcanivorax, we put several lines in Discussion, lines 451-453.

Specific comments; L115, how did you get the supernatant? Did you let the solids sedimentate first or did you use centrifugation?

**Response: we put the sample for 30 min and did not centrifuge. We modified the corresponding place.**

L175, give more information about

what you did with R and which packages you used.

**Response: yes, we inserted the linkage of the R scripts.**

L178, what is HMM

Response: HMM means hidden markov model. We have deleted it.

L203, what label did you use? Did you have a nonsense probe to check for unspecific labeling?

**Response:** actually we used E. coli to examine the probe as a negative control. We have inserted this in the new version.**

L223, MQ water?

**Response: we have spelled out.**

L234-251, check the figures. I thing the reference to Fig 2 should be Fig 3 and vice versa.

**Response: yes, modified.**

L255 ->, did you get any sequences from the blanks and if so,

**Response: The sequences in blanks were dimers of the primers for the amplification, which can be justified from the size of the smear. There were not any long amplified sequences from contaminated DNA.**

Results&Discussion, I would like to see some figures with the Alcanivorax genome and relevant metabolic maps. How does the Alcanivorax interact with the rest of the microbial community?

**Response: yes, we inserted a metabolic map for the Alcanivorax bacterium. Regarding the interaction, we inserted lines in Discussion, (lines 451-454). They are supposed to a convertor between alkanes and organic carbons and nitrogen.** Typos; L58, change 'were' to 'are' L81, do you mean 'found', not 'located'? L137, correct 'grounded' to 'ground' or 'homogenized'

Response: yes, all are corrected.

**Reviewer 2**

The manuscript submitted by Wang et al. describes the diversity of organisms in anhydrite crystals and further states a high abundance of Alcanivorax microbes. The authors see the specific community composition of the 750 years old crystals as indicator of an oxic oil-spilling benthic environment. Overall, the approach and results are highly interesting and definitely worth reporting in Biogeosciences. I, however, suggest revising and clarifying several parts of the manuscript before final publication.

General remarks:

While I do not doubt the results of the metagenomic investigation per se, I am honestly wondering what the meaning of the quantification of microbes based on traces of DNA is and how you could find intact cells being preserved for 750 years in crystals, which are still traceable using FISH.

Response: Since 2009, we have been working on this sediment layers for several years to answer the question regarding the elevated organic carbons in the ABS183 layer. The hypothesis is the preservation of cells in the crystals as we also revealed eukaryotic and archaeal sequences from metagenome. It was very surprising for us to find intact microbes in the crystals.

I am missing a detailed explanation and discussion on

that topic as also for me the microscopic pictures are not entirely satisfying. Response: we have to apologize for the quality of the pics. We selected these pictures to illustrate the presence of the microbes from hundreds of pictures. Although the quality is still not satisfying for you, the result is almost our best for readers. As we can understand, these microbes had been preserved for hundreds of years. It's difficult to find out well-preserved and well-labeled microbes.

Additionally

I would like to see some proof of the assembled genome and a metabolic map. Further, the conclusion on past oil-spilling is a bit too loosely connected to the other parts of the story, both in the abstract and the results and discussion parts. This could be done by introducing geochemical data earlier and by rephrasing some of the respective text parts. The references are not in Copernicus style.

Response: thanks for these comments. We have finished a new schematic map for the metabolism based on the gene profile. About the context of oil-spilling, we have rephrased the Abstract and inserted several lines in the Introduction (lines 139-157) to introduce the geochemical background in the oil-producing hydrothermal sediment.

Specific comments:

Abstract:

First sentence should be rephrased as a hypothesis.

**Response: yes**

1.33: which binning tool? Replace separated by assembled

**Response: we assembled the reads and then conducted binning of the contigs for a draft genome.**

1. 35. Remove 'the'

1.33-1.37: you are jumping back and forward between the metagenome and the cell identification

**Response: yes, these sentences are reorganized.**

1. 40 sentence is misplaced, here. Should probably be moved upwards.

**Response: yes**

1. 41-45: As written this seems to be highly speculative

**Response: yes, we have changed the tone.**

Introduction

L. 54: Repetition of previous sentence.

**Response: yes, deleted.**

1. 57: I don't understand this sentence.

**Response: rephrased!**

1. 59: replace 'prediction' by ' interpretation'

**Response: yes**

1. 60: replace 'have been' by 'are'

**Response: yes**

1. 63: The previous sentence already started with 'although'- replace.

**Response: yes, used 'but' now.**

1. 64: I would like to see a reference for this.

**Response: yes, two refs inserted.**

1. 69: replace 'become' by ' may get' and add a reference for this statement.

**Response: yes.**

1. 79: the bodies are the biomarkers? Actually lipids, DNA or also pigments may be used as biomarkers. Not sure whether the word 'body' is really the right expression dead microbial cells.

**Response: yes, it was replaced with 'evidence'. Line 118**

1. 81: similar to what?

**Response: changed. Line 120**

1. 95: 'Hence' is contradictory to 'probably'

**Response: yes.**

1. 97: Which other markers? Why probably? This would actually be the place to explain the geochemical background.

**Response: yes, we inserted several lines for the introduction.**

1. 100-104. This belongs to the methods part

**Response: this part was simplified.**

Material and methods

1. 110: A map of the sampling location would be beneficial.

**Response: we used a suppl figure S1.**

1. 138: Commonly, all abbreviations should be introduced when used for the first time.

**Response: yes.**

1. 146: this is a very low amount of DNA, how representative is this for community analysis? In how far is the quantification of OTUs and the genome identification trustworthy based on such low amounts of DNA? What about different degradation patterns of different organisms?

Response: We did not use up all the extraction. In total, there was not 1ng of DNA, so we have to use the amplification. The DNA might be used for many amplifications actually. The kit may allow for unbiased amplification of all sorts of DNA. We have a publication to support the linear amplification of microbial genomic DNA.

1. 150: Is this a random amplification?

Response: the theoretical basis and our test support the random amplification for the trace DNA.

1. 156: Which chemistry and protocol?

Some statistics on the sequencing (how many reads per run, how many of the identifiable, how many reads related to 16S rDNA, etc.) would be desirable.

Response: we should have put all the information. The sequencing was done in a service centre. We don't know how many sequences were produced in one run. Perhaps there were many samples in one runs.

1. 169: How?

Response: we inserted the command for this work (line2 239-240).

1. 169: Based on the low amount of DNA and amplification steps in between, I doubt the quantitative aspects of the analysis.

Response: we admit that this step is a brief statistics of the community. The abundance of the species could not be quantified in an accurate manner.

1.175: remove gap after ). Rstudio will not make any visualization without a prober script, so please provide details on your script. Also, provide the correct reference instead of the link to the homepage.

Response: yes. We inserted a linkage for the R scripts. See lines 247-248.

1. 181: It is unclear how the draft genome was assembled. What is the completeness? A genome plot would increase the credibility.

Response: the assembly work was introduced at line 236. The assembly made the reads into long contigs. Then the binning process grouped the contigs according

to their distinctive coverage levels and tetranucleotide frequencies. The completeness was assessed firstly by the number of single-copy genes. We have used Fig. S2 to illustrate the genomic alignment between Sk2 and ABS183. From the alignment, there are no notable gaps between the two genomes, indicating the high genomic completeness of the ABS183 strain.

1. 207: replace 'examine' by 'validate'

**Response: yes**

l. 223 MQ- please spell

**Response: yes**

I doubt that FISH produces a reliable signal if you only get 20pg of DNA out of the crystals.

Response: actually we obtained far more than 20pg of DNA. We concentrated the cells into a membrane, so that the microbes from many crystals were released and condensed on one membrane.

Results

1.232, 236, 238, 246, 249 : Check numbering of figures. Also, where do you refer to Fig. 5?

**Response: yes, there are mistakes in the figures. The figure 5 was described in line 364.**

1. 262: archaea, fungi without capital letters.

**Response: yes.**

1. 265: Why is that surprising?

**Response: deleted.**

1. 269 onwards: this needs a genome plot and a Kegg metabolic map.

**Response: we described the metabolic map of Figure 6. Lines 392-403.**

1. 278: I don't buy the quantitative aspect, here.

**Response: we deleted it.**

1. 346: postulate seems to be a bit strong for the line of evidence provided, here.

**Response: we used 'proposed' now.**

1. 346 ff: This information is what you need to put into the abstract to make it convincing **Response: this is a nice comment. See the second sentence in the Abstract.**

[revised manuscript text omitted]

Wang Yong 2016/9/28 8:56 PM **Deleted:** after mild hydrothermal activities

| Wang Yong 2016/9/28 9:10 PM                                                                                                                                          |  |  |  |  |
|----------------------------------------------------------------------------------------------------------------------------------------------------------------------|--|--|--|--|
| Deleted: a hydrothermal and hypersaline sediment core sampled from                                                                                            |  |  |  |  |
| Wang Yong 2016/9/28 9:07 PM                                                                                                                                          |  |  |  |  |
| Deleted: in the Red Sea                                                                                                                                              |  |  |  |  |
| Wang Yong 2016/9/28 11:28 AM                                                                                                                                         |  |  |  |  |
| Moved (insertion) [2]                                                                                                                                                |  |  |  |  |
| Wang Yong 2016/9/28 3:53 PM                                                                                                                                          |  |  |  |  |
| Deleted: alkane                                                                                                                                                      |  |  |  |  |
| Wang Yong 2016/9/28 11:25 AM                                                                                                                                         |  |  |  |  |
| Moved (insertion) [1]                                                                                                                                                |  |  |  |  |
| Wang Yong 2016/9/28 11:02 AM                                                                                                                                         |  |  |  |  |
| Deleted: Using a genome-binning method                                                                                                                        |  |  |  |  |
| Wang Yong 2016/9/28 11:05 AM                                                                                                                                         |  |  |  |  |
| Deleted: , a                                                                                                                                                         |  |  |  |  |
| Wang Yong 2016/9/28 3:56 PM                                                                                                                                          |  |  |  |  |
| Deleted: the Alcanivorax bacterium                                                                                                                     |  |  |  |  |
| Wang Yong 2016/9/28 11:02 AM                                                                                                                                         |  |  |  |  |
| Deleted: separated                                                                                                                                                   |  |  |  |  |
| Wang Yong 2016/9/28 11:23 AM                                                                                                                                         |  |  |  |  |
| Deleted: Phylogenetic and genomic analyses
revealed that this species was a close relative of
Alcanivorax backumensis Sk2. The draft genome      |  |  |  |  |
| Wang Yong 2016/9/28 11:22 AM                                                                                                                                         |  |  |  |  |
| Deleted: the                                                                                                                                                         |  |  |  |  |
| Wang Yong 2016/9/28 11:25 AM                                                                                                                                         |  |  |  |  |
| Moved up [1]: Fluorescence microscopy using 16S rRNA and marker gene probes revealed intact cells of the Alcanivorax bacterium in the crystals.        |  |  |  |  |
| Wang Yong 2016/9/28 11:28 AM                                                                                                                                         |  |  |  |  |
| Moved up [2]: The estimated age of the anhydrite layer was between 750-770 years, which might span the event of hydrothermal eruption into the benthic floor. |  |  |  |  |
| Wang Yong 2016/9/28 11:05 AM                                                                                                                                         |  |  |  |  |
| Deleted: support                                                                                                                                                     |  |  |  |  |
| Wang Yong 2016/9/28 11:06 AM                                                                                                                                         |  |  |  |  |
|                                                                                                                                                                      |  |  |  |  |

**72 1. Introduction**

85

73 Deep-sea sediment is among the least explored biospheres on Earth. Indigenous microbes 74 differ vastly in community composition and metabolic spectra at different depths and sites 75 sites (Orcutt et al., 2011; Teske and Sorensen, 2007). The distribution of microbes in 76 subsuperficial sediments is determined by the porosity, nutrient availability and geochemical 77 geochemical conditions of the sediment (Parkes et al., 2000; Webster et al., 2006). In return, 78 return, genomic features and the community composition of the indigenous microbial 79 inhabitants may reflect the *in situ* conditions and serve as biomarkers containing the 80 geochemical indicators. However, most of the biomarkers cannot be well preserved and will 81 will be degraded by biological and abiological activities. Although lipids and other organic organic carbons present in some minerals allow the interpretation of microbial activities to 82 to some extent (Brocks et al., 2005), the original metabolic activities are difficult to retrieve 83 84 retrieve in a comprehensive and precise manner.

86 Most of the dead microbes are damaged during the sedimentation process, but some can be 87 be maintained in almost their original shape (Taher, 2014; Benison et al., 2008). Evaporites, 88 Evaporites, which mostly consist of halite and anhydrite (CaSO4) or gypsum (CaSO4·2H2O, 89 (CaSO4·2H2O, temperature <38°C (Hill, 1937)), are common microbialites with 90 accretionary organosedimentary structures (Dupraz et al., 2011). Numerous dead bacteria, 91 algae and metazoans have been detected in gypsum granules (Petrash et al., 2012; Trichet et 92 et al., 2001); bacterial mats growing on evaporites may become trapped and constitute much 93 much larger microbialites (Babel, 2004). Consequently, microbial inhabitants on the benthic 94 benthic surface may get trapped in the evaporites (Benison et al., 2008). Anhydrite facies are 95 are not found throughout deep-sea sediments. They usually form around hydrothermal vents 96 vents in deep-sea environments (Jannasch and Mottl, 1985). A strong deep-sea volcanic 97 eruption may break the crustal basalts, resulting in a drastic emission of hydrothermal gases 98 gases followed by the crystallization of anhydrites and the deposition of metal sulfides 99 (Kristall et al., 2006). An alternative model is that mild hydrothermal activities lead to a slow 100 slow influx of solutions into the overlying sediment at temperatures in the sub-seafloor 101 ranging from 20-100°C. This process also results in the formation of crystalline anhydrites 102 anhydrites in veins and around warm vents (Jannasch and Mottl, 1985). The latter process

**Wang Yong 2016/9/28 3:59 PM**

| 1 | Wang Yong 2016/9/28 4:03 PM                                                                                                                       |  |  |  |
|---|---------------------------------------------------------------------------------------------------------------------------------------------------|--|--|--|
|   | Deleted: dead microbes                                                                                                                            |  |  |  |
| 4 | Wang Yong 2016/9/28 4:05 PM                                                                                                                       |  |  |  |
|   | Deleted: Most of the biological markers containing containing the geochemical indicators awere lost due to lack of preservation processes. |  |  |  |
| Ϊ | Wang Yong 2016/9/28 4:05 PM                                                                                                                       |  |  |  |
|   | Deleted: dict                                                                                                                                     |  |  |  |
| Ϊ | Wang Yong 2016/9/28 4:06 PM                                                                                                                       |  |  |  |
|   | Deleted: have been                                                                                                                                |  |  |  |
| 1 | Wang Yong 2016/9/28 4:07 PM                                                                                                                       |  |  |  |
|   | Deleted: Although m                                                                                                                               |  |  |  |
|   | Wang Yong 2016/9/28 4:26 PM                                                                                                                       |  |  |  |
|   | Deleted: likely                                                                                                                                   |  |  |  |

Wang Yong 2016/9/28 4:31 PM Deleted: become Wang Yong 2016/9/28 4:31 PM Deleted: evaporites

[revised manuscript text omitted]

Wang Yong 2016/9/29 5:44 PM Deleted: +

Wang Yong 2016/9/29 5:44 PM Deleted: ica et Wang Yong 2016/9/29 5:44 PM Deleted: ochimica Wang Yong 2016/9/29 5:44 PM Deleted: ards in Wang Yong 2016/9/29 5:44 PM Deleted: g Wang Yong 2016/9/29 5:45 PM Deleted: ic Wang Yong 2016/9/29 5:45 PM Deleted: sciences Wang Yong 2016/9/29 5:45 PM Deleted: Standards in genomic sciences Wang Yong 2016/9/29 5:45 PM Deleted: m Wang Yong 2016/9/29 5:45 PM Deleted: ogy ecology

Wang Yong 2016/9/29 5:46 PM Deleted: 10.1130/0091-7613(1995)023<0543:nsithb>2.3.co;

- 605 Brocks, J. J., Love, G. D., Summons, R. E., Knoll, A. H., Logan, G. A., and Bowden, S. 606 A .: Biomarker evidence for green and purple sulphur bacteria in a stratified 607 Palaeoproterozoic sea, Nature, 437, 866-870, 2005. 608 Cameron, D., Willett, M., and Hammer, L.: Distribution of organic carbon in the 609 Berkeley Pit lake, Butte, Montana, Mine Water Environ, 25, 93-99, 610 10.1007/s10230-006-0116-4, 2006. Dupraz, C., Reid, R. P., and Visscher, P. T.: Microbialites, modern, in: Encyclopaedia of 611 612 Geobiology, edited by: Reitner, V., and Thiel, J., Springer, Heidelberg, 617-635, 2011. 613 Girdler, R. W.: A review of Red Sea heat flow, Phil Trans Roy Soc Lon A, 267, 191-203, 1970. 614 615 Goris, J., Konstantinidis, K. T., Klappenbach, J. A., Coenye, T., Vandamme, P., and Tiedje, J. M .: DNA-DNA hybridization values and their relationship to whole-genome 616 617 sequence similarities, Int J Syst Evol Microbiol, 57, 81-91, 2007. 618 Gough, H. L., and Stahl, D. A.: Optimization of direct cell counting in sediment. , J 619 Microbiol Methods, 52, 39-46, 2003. 620 Gutierrez, T., Singleton, D. R., Berry, D., Yang, T., Aitken, M. D., and Teske, A.: 621 Hydrocarbon-degrading bacteria enriched by the Deepwater Horizon oil spill 622 identified by cultivation and DNA-SIP, ISME J, 7, 2091-2104, 10.1038/ismej.2013.98, 623 2013. 624 Hill, A. E.: The transition temperature of gypsum to anhydrite, J Am Chem Soc, 59, 625 2242-2244, 10.1021/ja01290a039, 1937. 626 Huang, Y., Gilna, P., and Li, W.: Identification of ribosomal RNA genes in metagenomic 627 fragments, Bioinformatics, 25, 1338-1340, 2009. Hyatt, D., Chen, G. L., LoCascio, P. F., Land, M. L., Larimer, F. W., and Hauser, L. J.: 628 629 Prodigal: prokaryotic gene recognition and translation initiation site identification, 630 Bmc Bioinformatics, 11, Artn 119 631 Doi 10.1186/1471-2105-11-119, 2010. 632 Jannasch, H. W., and Mottl, M. J.: Geomicrobiology of deep-sea hydrothermal vents, 633 Science, 229, 717-725, 10.1126/science.229.4715.717, 1985. 634 Kanehisa, M., Goto, S., Sato, Y., Furumichi, M., and Tanabe, M.: KEGG for integration 635 and interpretation of large-scale molecular data sets, Nucleic Acids Res, 40, D109-114, 636 10.1093/nar/gkr988
  - 637 gkr988 [pii], 2012.
  - 638 Kim, O. S., Cho, Y. J., Lee, K., Yoon, S. H., Kim, M., Na, H., Park, S. C., Jeon, Y. S.,
  - 639 Lee, J. H., Yi, H., Won, S., and Chun, J.: Introducing EzTaxon-e: a prokaryotic 16S
  - rRNA gene sequence database with phylotypes that represent uncultured species, Int J
     Syst Evol Microbiol, 62, 716-721, 10.1099/ijs.0.038075-0, 2012.
  - 641 Syst Evol Microbiol, 62, 716-721, 10.1099/ijs.0.038075-0, 2012.
  - Klein, A. N., Frigon, D., and Raskin, L.: Populations related to Alkanindiges, a novel
    genus containing obligate alkane degraders, are implicated in biological foaming in
    activated sludge systems, Environ Microbiol, 9, 1898-1912,
  - 645 10.1111/j.1462-2920.2007.01307.x, 2007.
  - Klinkhammer, G. P., and Lambert, C. E.: Preservation of organic matter during salinity
     excursions, Nature, 339, 271-274, 1989.
  - 648 Kristall, B., Kelley, D. S., Hannington, M. D., and Delaney, J. R.: Growth history of a
  - diffusely venting sulfide structure from the Juan de Fuca Ridge: A petrological and

Wang Yong 2016/9/29 5:47 PM Deleted: International journal of systematic and evolutionary microbiology

http://www.nature.com/nature/journal/v437/n7060/ suppinfo/nature04068\_S1.html

Wang Yong 2016/9/29 5:47 PM

- 657 geochemical study, Geochem. Geophys. Geosyst., 7, Q07001, 10.1029/2005gc001166, 658 2006.
- 659 Li, H., Handsaker, B., Wysoker, A., Fennell, T., Ruan, J., Homer, N., Marth, G., Abecasis,
- G., and Durbin, R.: The sequence alignment/map format and SAMtools, 660
- 661 Bioinformatics, 25, 2078-2079, DOI 10.1093/bioinformatics/btp352, 2009.
- 662 Ludwig, W., Strunk, O., Westram, R., Richter, L., Meier, H., Yadhukumar, Buchner, A.,
- Lai, T., Steppi, S., Jobb, G., Forster, W., Brettske, I., Gerber, S., Ginhart, A. W., Gross, 663
- 664 O., Grumann, S., Hermann, S., Jost, R., Konig, A., Liss, T., Lussmann, R., May, M.,
- Nonhoff, B., Reichel, B., Strehlow, R., Stamatakis, A., Stuckmann, N., Vilbig, A., 665
- Lenke, M., Ludwig, T., Bode, A., and Schleifer, K.-H.: ARB: a software environment 666
- 667 for sequence data, Nucl. Acids Res., 32, 1363-1371, 10.1093/nar/gkh293, 2004.
- 668 Mahmood, Q., Hu, B., Cai, J., Zheng, P., Azim, M. R., Jilani, G., and Islam, E.: Isolation 669 of Ochrobactrum sp.QZ2 from sulfide and nitrite treatment system, J Hazardous
- 670 Materials, 165, 558-565, http://dx.doi.org/10.1016/j.jhazmat.2008.10.021, 2009.
- Meier-Kolthoff, J. P., Auch, A. F., Klenk, H. P., and Goker, M.: Genome sequence-based 671 672 species delimitation with confidence intervals and improved distance functions, BMC 673 Bioinfo, 14, 60, 10.1186/1471-2105-14-60, 2013.
- Missack, E., Stoffers, P., and El Goresy, A.: Mineralogy, paragenesis, and phases 674 relations of copper iron sulfides in the Atlantis II Deep, Red Sea, Min. Deposita, 24, 675
- 676 82-91, 1989. 677 Nurk, S., Bankevich, A., Antipov, D., Gurevich, A. A., Korobevnikov, A., Lapidus, A.,
- 678 Prjibelski, A. D., Pvshkin, A., Sirotkin, A., Sirotkin, Y., Stepanauskas, R., Clingenpeel, 679 S. R., Woyke, T., McLean, J. S., Lasken, R., Tesler, G., Alekseyev, M. A., and
- 680 Pevzner, P. A.: Assembling single-cell genomes and mini-metagenomes from chimeric MDA products, J Comput Biol, 20, 714-737, 10.1089/cmb.2013.0084, 2013. 681
- Orcutt, B. N., Sylvan, J. B., Knab, N. J., and Edwards, K. J.: Microbial ecology of the 682 683 dark ocean above, at, and below the seafloor, Microbiol Mol Biol Rev, 75, 361-422, 684 Doi 10.1128/Mmbr.00039-10, 2011.
- Oudin, E., Thisse, Y., and Ramboz, C.: Fluid inclusion and mineralogical evidence for 685 686 high temperature saline hydrothermal circulation in the Red Sea metalliferous 687 sediments: preliminary results, Mar. Mining, 5, 3-31, 1984.
- 688 Overbeek, R., Begley, T., Butler, R. M., Choudhuri, J. V., Chuang, H. Y., Cohoon, M., de
- 689 Crecy-Lagard, V., Diaz, N., Disz, T., Edwards, R., Fonstein, M., Frank, E. D., Gerdes,
- 690 S., Glass, E. M., Goesmann, A., Hanson, A., Iwata-Reuyl, D., Jensen, R., Jamshidi, N.,
- Krause, L., Kubal, M., Larsen, N., Linke, B., McHardy, A. C., Meyer, F., Neuweger, 691
- H., Olsen, G., Olson, R., Osterman, A., Portnoy, V., Pusch, G. D., Rodionov, D. A., 692
- 693 Ruckert, C., Steiner, J., Stevens, R., Thiele, I., Vassieva, O., Ye, Y., Zagnitko, O., and 694
- Vonstein, V.: The subsystems approach to genome annotation and its use in the project 695 to annotate 1000 genomes, Nucl Acids Res, 33, 5691-5702, 33/17/5691 [pii]
- 696 10.1093/nar/gki866, 2005.
- 697 Parkes, R. J., Cragg, B. A., and Wellsbury, P.: Recent studies on bacterial populations
- 698
- and processes in subseafloor sediments: A review, Hydrogeol J. 8, 11-28,
- 699 10.1007/pl00010971, 2000.
- 700 Party, T. S. S.: Red Sea: Site 226, in: Deep Sea Drilling Project, 595-600, 1974.

Wang Yong 2016/9/29 5:53 PM Formatted: Default Paragraph Font, English (UK), Check spelling and grammar

Wang Yong 2016/9/29 5:47 PM Deleted: b Wang Yong 2016/9/29 5:48 PM Deleted: rmatics

Wang Yong 2016/9/29 5:48 PM Deleted: eic

Wang Yong 2016/9/29 5:48 PM Deleted: ogy Wang Yong 2016/9/29 5:48 PM Deleted: ournal

- Pernthaler, A., Pernthaler, J., and Amann, R.: Fluorescence in situ hybridization and
   catalyzed reporter deposition for the identification of marine bacteria, Appl Environ
- 708 Microb, 68, 3094-3101, 2002.
- Petrash, D. A., Gingras, M. K., Lalonde, S. V., Orange, F., Pecoits, E., and Konhauser, K.
  O.: Dynamic controls on accretion and lithification of modern gypsum-dominated
  thrombolites, Los Roques, Venezuela, Sedi Geol, 245-246, 29-47, 2012.
- 711 Infolhoontes, Los Roques, Venezueia, Sedi Geoi, 243-240, 29-47, 2012.
   712 Quast, C., Pruesse, E., Yilmaz, P., Gerken, J., Schweer, T., Yarza, P., Peplies, J., and
- Glöckner, F. O.: The SILVA ribosomal RNA gene database project: improved data
- 714 processing and web-based tools, Nucl Acids Res, 41, D590-D596,
- 715 10.1093/nar/gks1219, 2013.
- Ramboz, C., Oudin, E., and Thisse, Y.: Geyser-type discharge in Atlantis II Deep, Red
  Sea: evidence of boiling from fluid inclusions in epigenetic anhydrite, Can Mineral, 26,
  765-786, 1988.
- 719 Reimer, P. J., Baillie, M. G. L., Bard, E., Bayliss, A., Beck, J. W., Blackwell, P. G.,
- Ramsey, C. B., Buck, C. E., Burr, G. S., Edwards, R. L., Friedrich, M., Grootes, P. M.,
  Guilderson, T. P., Hajdas, I., Heaton, T. J., Hogg, A. G., Hughen, K. A., Kaiser, K. F.,
- 722 Kromer, B., McCormac, F. G., Manning, S. W., Reimer, R. W., Richards, D. A.,
- Southon, J. R., Talamo, S., Turney, C. S. M., van der Plicht, J., and Weyhenmeyer, C.
  E.: IntCal09 and Marine09 radiocarbon age calibration curves, 0-50,000 yeats cal BP,
- 725 Radiocarbon, 51, 1111-1150, 2009.
- Richter, M., and Rossello-Mora, R.: Shifting the genomic gold standard for the
  prokaryotic species definition, Proc Natl Acad Sci U S A, 106, 19126-19131,
  10.1073/pnas.0906412106, 2009.
- Sabirova, J. S., Becker, A., Lunsdorf, H., Nicaud, J. M., Timmis, K. N., and Golyshin, P.
  N.: Transcriptional profiling of the marine oil-degrading bacterium Alcanivorax
  borkumensis during growth on n-alkanes, FEMS Microbiol Lett, 319, 160-168,
  10.1111/j.1574-6968.2011.02279.x, 2011.
- Schardt, C.: Hydrothermal fluid migration and brine pool formation in the Red Sea: the
  Atlantis II Deep, Mineralium Deposita, 51, 89-111, 10.1007/s00126-015-0583-2,
  2016.
- 736 Schneiker, S., Martins dos Santos, V. A., Bartels, D., Bekel, T., Brecht, M., Buhrmester,
- J., Chernikova, T. N., Denaro, R., Ferrer, M., Gertler, C., Goesmann, A., Golyshina, O.
- V., Kaminski, F., Khachane, A. N., Lang, S., Linke, B., McHardy, A. C., Meyer, F.,
   Nechitaylo, T., Puhler, A., Regenhardt, D., Rupp, O., Sabirova, J. S., Selbitschka, W.,
- 739 Nechnaylo, 1., Punier, A., Regennardt, D., Rupp, O., Sabirova, J. S., Seibischka, 740 Yakimov, M. M., Timmis, K. N., Vorholter, F. J., Weidner, S., Kaiser, O., and
- 740 Fakinov, M. M., Finnins, K. N., Vollolet, F. J., Weldier, S., Kalser, O., and 741 Golyshin, P. N.: Genome sequence of the ubiquitous hydrocarbon-degrading marine
- bacterium Alcanivorax borkumensis, Nature Bjotech, 24, 997-1004, 10.1038/nbt1232,
   2006
- Simoneit, B. R. T.: Petroleum generation submarine hydrothermal systems: An update,
   Can Mineral, 26, 827-840, 1988.
- Swallow, J. C., and Crease, J.: Hot salty water at the bottom of the Red Sea, Nature, 205, 165-166, 1965.
- 748 Taher, A. G.: Microbially induced sedimentary structures in evaporite-siliciclastic
- sediments of Ras Gemsa sabkha, Red Sea Coast, Egypt, J Adv Res, 5, 577-586,
- 750 http://dx.doi.org/10.1016/j.jare.2013.07.009, 2014.

Wang Yong 2016/9/29 5:48 PM Deleted: b Wang Yong 2016/9/29 5:49 PM Deleted: nology

Wang Yong 2016/9/29 5:49 PM Formatted: Default Paragraph Font, English (UK), Check spelling and grammar (

- Tamura, K., Peterson, D., Peterson, N., Stecher, G., Nei, M., and Kumar, S.: MEGA5:
   molecular evolutionary genetics analysis using maximum likelihood, evolutionary
   distance, and maximum parsimony methods, Mol Biol Evol, 28, 2731-2739, 2011.
- Teske, A., and Sorensen, K. B.: Uncultured archaea in deep marine subsurface sediments:
  have we caught them all?, ISME J, 2, 3-18, 2007.
- Trichet, J., Défarge, C., Tribble, J., Tribble, G. W., and Sansone, F. J.: Christmas Island
  lagoonal lakes, models for the deposition of carbonate-evaporite-organic laminated
  sediments, Sedimentary Geol, 140, 177-189, 2001.
- Wang, W., Wang, L., and Shao, Z.: Diversity and abundance of oil-degrading bacteria and alkane hydroxylase (alkB) genes in the subtropical seawater of Xiamen Island, Microb Ecol, 60, 429-439, 10.1007/s00248-010-9724-4, 2010.
- Wang, Y., Yang, J., Lee, O. O., Dash, S., Lau, S. C. K., Al-Suwailem, A., Wong, T. Y. H.,
  Danchin, A., and Qian, P.-Y.: Hydrothermally generated aromatic compounds are
  consumed by bacteria colonizing in Atlantis II Deep of the Red Sea, ISME J, 5, 1652–
  1659, 2011.
- Wang, Y., Li, J. T., He, L. S., Yang, B., Gao, Z. M., Cao, H. L., Batang, Z., Al-Suwailem,
  A., and Qian, P. Y.: Zonation of microbial communities by a hydrothermal mound in
  the Atlantis II Deep (the Red Sea), PLoS ONE, 10, e0140766,
- 771 10.1371/journal.pone.0140766, 2015.
- Wang, Y., Gao, Z. M., Xu, Y., Li, G. Y., He, L. S., and Qian, P. Y.: An evaluation of
  multiple annealing and looping based genome amplification using a synthetic bacterial
  community, ACTA Oceanol Sin, 35, 131-136, 2016.
- Wayne, L. G., Brenner, D. J., Colwell, R. R., Grimont, P. A. D., Kandler, O., Krichevsky,
  M. I., Moore, L. H., Moore, W. E. C., Murray, R. G. E., Stackebrandt, E., Starr, M. P.,
  and Trüper, H. G.: Report of the ad hoc committee on reconciliation of approaches to
  bacterial systematics, Int J Syst Bacteriol, 37, 463-464, 1987.
- Webster, G., John Parkes, R., Cragg, B. A., Newberry, C. J., Weightman, A. J., and Fry, J.
  C.: Prokaryotic community composition and biogeochemical processes in deep
  subseafloor sediments from the Peru Margin, FEMS Microbiol Ecol, 58, 65-85,
  10.1111/j.1574-6941.2006.00147.x, 2006.
- Wu, Y., He, T., Zhong, M., Zhang, Y., Li, E., Huang, T., and Hu, Z.: Isolation of marine
   benzo[a]pyrene-degrading Ochrobactrum sp. BAP5 and proteins characterization, J

785 Environ Sci, 21, 1446-1451, 2009.

- Yakimov, M. M., Timmis, K. N., and Golyshin, P. N.: Obligate oil-degrading marine
  bacteria, Curr Opin Biotech, 18, 257-266, 2007.
- Zierenberg, R. A., and Shanks, W. C.: Mineralogy and geochemistry of epigenetic
  features in metalliferous sediment, Atlantis II Deep, Red Sea, Econ Geol, 78, 57-72,
  10.2113/gsecongeo.78.1.57, 1983.
- Zong, C., Lu, S., Chapman, A. R., and Xie, X. S.: Genome-wide detection of
  single-nucleotide and copy-number variations of a single human cell, Science, 338,
  1622-1626, 10.1126/science.1229164, 2012.
- Zu, L., Xiong, J., Li, G., Fang, Y., and An, T.: Concurrent degradation of
   tetrabromobisphenol A by Ochrobactrum sp. T under aerobic condition and estrogenic
   transition during these processes, Ecot Eviron Safety, 104, 220-225,
- 797 10.1016/j.ecoenv.2014.03.015, 2014.
- 797 10.1016/j.ecoenv.2014.03.015, 798

Wang Yong 2016/9/29 5:49 PM Deleted: emational journal of Wang Yong 2016/9/29 5:50 PM Deleted: systematic bacteriology

Wang Yong 2016/9/29 5:50 PM Deleted: http://dx.doi.org/10.1016/S1001-0742(08)62438-9 Wang Yong 2016/9/29 5:50 PM Formatted: Default Paragraph Font, English (UK), Check spelling and grammar Wang Yong 2016/9/29 5:50 PM Deleted: http://dx.doi.org/10.1016/j.copbio.2007.04.006 Wang Yong 2016/9/29 5:50 PM Deleted: omic Wang Yong 2016/9/29 5:50 PM Deleted: ogy Wang Yong 2016/9/29 5:50 PM Deleted: oxicology and Wang Yong 2016/9/29 5:51 PM Deleted: environmental safety

**809 Data Accessibility**

- 810 Illumina raw data will be accessible under SRA356974 in the NCBI SRA database. B.
- 811 borkumensis ABS183 genome was deposited in the NCBI under BioProject
- 812 LKAP00000000 and will be public on October 31, 2016.

813

814

815 **Table 1.** Age estimates of the sediment layers

| Layer (cm) | Age (year) | Age error (year) |
|------------|------------|------------------|
| 3-6        | 320        | 25               |
| 21-24      | 475        | 35               |
| 45-48      | 490        | 30               |
| 90-93      | 500        | 25               |
| 129-132    | 560        | 35               |
| 153-156    | 750        | 30               |
| 198-201    | 770        | 30               |
| 222-225    | 880        | 30               |

816

817 Eight sediment layers were selected for the age estimates using radioisotope  ${}^{14}C$  of G.

818 sacculifer collected from the respective layers. The age was corrected by the 400-year

819 reservoir age with an error range.

820

**822 Figures**

**823 Figure 1. Anhydrite crystals and genome binning.**

- 824 Anhydrite crystals in a Petri dish (90 mm in diameter) (A) were used for DNA extraction.
- 825 The amplified genomic DNA was sequenced and then reassembled. Based on the G+C
- 826 content and read coverage, the binned contigs with high coverage levels (B) were
- 827 selected for examination of the tetranucleotide frequency consistency in the PCA analysis
- 828 (C).

829

830

**831 Figure 2, Grain size and age of selected layers**

The percentages of the small particles ( $

---

## Referee Report (RR1)

Wang et al. - Review no. 2

After revision, the manuscript of Wang et al. increased a lot in readability, most of my suggestions and the ones from reviewer 1 were included into the revised manuscript. Most formal corrections and requests have been addressed.

Still, there are details that have to be addressed, such as the sequencing statistics. This is something that is visible from the raw sequencing files and can hardly been unseen. It is not credible that the authors do not have access to these data, while stating that they did the analysis on their own. I would thus like to see the sequence quality files as part of the fastq files or the sequence raw data. In addition, I have to insist on the information of the sequencing chemistry and protocol, as there are contradicting information even on the type of sequencer between what is given in the methods part and in the NCBI sequence read archive.

Further, I still do not believe the quantification of microbes based on traces of DNA, the quantitative aspect of the metagenomics dataset is not credible based on a not representative amount of DNA, which has likely been subject to different degradation processes. How much DNA has been isolated in total?

Specific comments (track change version):

l. 102 remove second 'anhydrites', there are several repeated words down to line 128, please remove

l. 239 Rephrase: Using the classify.seqs command as part of the Mothur software package. Add a reference instead of a link

l. 247 replace 'in' by 'from'

l. 393 I don't see genes indicating nitrogen fixation, here

l. 455 'propose'

l. 810 ff 'A. borkumensis',… genome was deposited at the NCBI database, Bioproject…

---

## Author Response (AR2)

Wang et al. - Review no. 2

After revision, the manuscript of Wang et al. increased a lot in readability, most of my suggestions and the ones from reviewer 1 were included into the revised manuscript. Most formal corrections and requests have been addressed.

Still, there are details that have to be addressed, such as the sequencing statistics. This is something that is visible from the raw sequencing files and can hardly been unseen. It is not credible that the authors do not have access to these data, while stating that they did the analysis on their own. I would thus like to see the sequence quality files as part of the fastq files or the sequence raw data. In addition, I have to insist on the information of the sequencing chemistry and protocol, as there are contradicting information even on the type of sequencer between what is given in the methods part and in the NCBI sequence read archive.

**Response: thanks for this criticism. We split our data and made a statistic table using the sequencing data. Table S1 shows the details. There was a typo in the previous version. We used Illumina 2000, instead of Illumina 2500. Sorry for the mistake. The sequencing data for the anhydrite amplified DNA were 1.8Gb although we requested 2Gb as shown in the NCBI SRA. After quality control, 1.6Gb was left. After removal of MALBAC primers, we obtained 0.9Gb clean data for assembly and genomic binning. See 275-277 for the description of the data.**

Further, I still do not believe the quantification of microbes based on traces of DNA, the quantitative aspect of the metagenomics dataset is not credible based on a not representative amount of DNA, which has likely been subject to different degradation processes. How much DNA has been isolated in total?

**Response: There was a total of 500pg of raw DNA using 20g crystals. The quantification was done with a Quant-iT PicoGreen kit (Invitrogen, USA) (line 159). At present, DNA less than 1ng could not be used for library preparation for Illumina. The MALBAC can amplify DNA into fragments of small sizes between 400-2000bp, so degradation of DNA won't notably affect linear amplification of genomic DNA. The linear amplification of MALBAC has been demonstrated in our publication Acta Oceanol. Sin., 2016, Vol. 35, No. 2, P. 131–136 in the reference list. We admit that quantification of microbes was not accurate based on the low amount of DNA. At least, our result showed the compositional discrepancy between the communities in the crystals and the control.**

Specific comments (track change version):

l. 102 remove second 'anhydrites', there are several repeated words down to line 128, please remove

**Response: This was caused by PDF maker. I cannot find the problem in the final version.**

l. 239 Rephrase: Using the classify.seqs command as part of the Mothur software package. Add a reference instead of a link

**Response: yes. Done.**

l. 247 replace 'in' by 'from'

**Response: revised!**

l. 393 I don't see genes indicating nitrogen fixation, here

**Response: Sorry it was a typo. We revised the sentences in lines 325-328. Ammonia might be imported and assimilated into glutamate as depicted in figure 6.**

l. 455 'propose'

**Response: yes**

810: **Response: the sentence was revised!**

---

## Author Response (AR3)

The authors now provided the missing data and did overall a good job in revising their manuscript. I thus have only technical comments left:
**Response: thanks for the careful review!**

l.193: Give a reference for the R program
**Response: yes.**

l. 197: Give a reference for hmmsearch
**Response: yes.**

l. 204: Give a reference for KEGG
**Response: there is a reference for KEGG database before the sentence that mentioned KEGG website. For the website, we just used a URL link.**

l. 211 Give a reference for GGDC
**Response: There are three references for it. We moved them forward to be closer to 'GGDC'.**

l. 229 Give a reference for first isolation of A. borkumensis
**Response: yes, we found the first publication for Alcanivorax borkumensis Sk2.**

l. 231: company, location

**Response: yes.**

l. 234: spell out PBS
**Response: yes.**

l. 245: DAPI: company, location
**Response: yes.**

Figures:

Figure 1: labels on Fig. 1b, c are too small, legend in 1b is too small
**Response: the figure has been modified!**
Figure 4: there are some shadows around the circles, maybe a higher resolution could help, here
**Response: The shadows have been removed.**
Figure S2: The comparison was conducted on the ACT website using what tool?
**Response: the legend was modified as "The comparison was**

**conducted using Artemis Comparison Tool in the WebACT (webact.org)."**

Table S1: Replace heading by ' Sequencing statistics'
**Response: yes.**